# Multi-condensate state as a functional strategy to optimize the cell signaling output

Aniruddha Chattaraj ⬥[1] & Eugene I. Shakhnovich ⬥[1] ✉

The existence of multiple biomolecular condensates inside living cells is a peculiar phenomenon not compatible with the predictions of equilibrium statistical mechanics. In this work, we address the problem of multiple condensates state (MCS) from a functional perspective. We combine Langevin dynamics, reaction-diffusion simulation, and dynamical systems theory to demonstrate that MCS can indeed be a function optimization strategy. Using Arp2/3 mediated actin nucleation pathway as an example, we show that actin polymerization is maximum at an optimal number of condensates. For a fixed amount of Arp2/3, MCS produces a greater response compared to its single condensate counterpart. Our analysis reveals the functional significance of the condensate size distribution which can be mapped to the recent experimental findings. Given the spatial heterogeneity within condensates and non-linear nature of intracellular networks, we envision MCS to be a generic functional solution, so that structures of network motifs may have evolved to accommodate such configurations.

Spatiotemporal control of biochemical events enables living cells to respond to changes in the environment and other perturbations. orchestration of cellular biochemistry in time and space is achieved via multiple mechanisms. A key aspect of this organization is the presence of distinct subcellular compartments, such as the nucleus, mitochondria, and Golgi body, each dedicated to specific functions[1]. For example, the nucleus is responsible for storing DNA and facilitating mRNA production, while the mitochondria act as the cellular powerhouse, regulating energy metabolism. This task-specific compartmentalization of subcellular space is arguably the most significant step in the evolution of complex life forms[1].

Over the last decade, another class of intracellular structures, known as biomolecular condensates[2] that lack a surrounding membrane, has emerged. Clustering of weakly interacting multivalent proteins and nucleic acids underlies the condensate formation[3–7]. These proteins have multiple interaction "sites" or "domains" which can form transient physical crosslinks ("bonds") with their partner molecules. These interaction motifs are usually referred to as "stickers", which are interspersed with "spacer" regions[6,8]. The extent of crosslinking increases with concentration and beyond a system-specific threshold concentration, large multi-molecular clusters emerge as phase-separated droplets. Theories from polymer sciences

greatly shape our understanding of the underlying physics of biological condensates[8–10]. However, one phenomenon remains enigmatic. Classical theory predicts that, after phase transition, the system should form one large droplet coexisting with the soluble phase. But, till now, the majority of cellular and in-vitro experiments revealed a multi-condensate state (MCS) where the liquid droplets dynamically exchange components and never fully coalesce to become one large droplet within the experimental timespan.

We have previously proposed that the MCS is a dynamically arrested metastable state[7] where the interplay of timescales of diffusion and internal relaxation within emerging clusters determine the metastability of condensates and their size distribution. Inter-particle bonding kinetics are also shown to be important for guiding self-assembly pathways in colloidal models[11]. In the current work, we approach the problem of MCS from a functional perspective. We ask whether a state with multiple droplets is functionally more useful than its single droplet counterpart? To answer that question, we consider an experimentally well-characterized system of Arp2/3 mediated actin nucleation pathway[4,12–14].

Condensate-forming biomolecules can be broadly classified into two classes—"scaffolds" and "clients"[15]. Scaffolds are necessary core components that control the size and composition of the condensates,

[1]Department of Chemistry and Chemical Biology, Harvard University, Cambridge, MA 02138, USA. ✉e-mail: shakhnovich@chemistry.harvard.edu

while clients are selectively recruited to these condensates, presumably serving to establish microenvironments that foster specific biochemical reactions. Furthermore, an emerging paradigm in the field underscores the presence of spatial heterogeneity within these condensates, with distinct molecular species occupying discrete regions - a structural hallmark observed in mesoscopic bodies such as the Nucleolus[16], P-granules[17,18], Stress granules[19], Paraspeckles[20], and Nanog condensates[21]. However, it is not yet clear whether such spatial assembly has any functional relevance. The topological relationship between molecular structures (sequence of sticker and spacer, for example) and the mesoscopic body (spatially heterogenous condensates) they assemble into is an intriguing evolutionary question.

In this study, we consider a minimal-component system that is computationally tractable and experimentally well-studied. Our system consists of four upstream components (Nephrin, Nck, NWASP, and Arp2/3, Fig. 1A) of the actin nucleation pathway[14,22–25]. Polymerization of the cytoskeletal protein actin is an important process that cells utilize to respond to their environment. The dynamic equilibrium between monomeric (Globular, G) and polymeric (Filament, F) actin is regulated by a wide variety of proteins. Arp2/3 complex is a nucleation-promoting factor[26] that promotes the formation of the branched F-actin network. In resting cells, Arp2/3 remains in an "off"

state. Given an extracellular signal, membrane-bound receptor, Nephrin forms a cluster and binds to an adapter protein, Nck. Nck recruits NWASP, which, in turn, activates the Arp2/3 complex and initiates the F-actin branching. The active Arp2/3 sits on the side of an existing filament ("mother filament") and creates a branched filament ("daughter filament") along a 70° angular direction. The extent of the branched F-actin network connects the cell's ability to effectively respond to the external signal. In a series of experiments[12,13], Rosen and colleagues showed that Nephrin, Nck and NWASP form condensates, both in test-tube and cells. These condensates recruit Arp2/3 and promote local production of F-actin. Therefore, we consider modeling this system as a case study to unravel the functional picture of biological condensates.

Computational models provide useful quantitative and mechanistic insights that, otherwise, may not be accessible via experiments. However, these models are limited to a specific spatiotemporal scale, although any biological problem inherently encompasses multiple scales. To overcome this limitation, we employed a multi-scale simulation strategy. First, we used Langevin Dynamics simulation to probe the structural organization of the condensates. Informed by these finer-resolution simulations, we created a reaction-diffusion model to couple the effect of condensate geometry with protein diffusion.

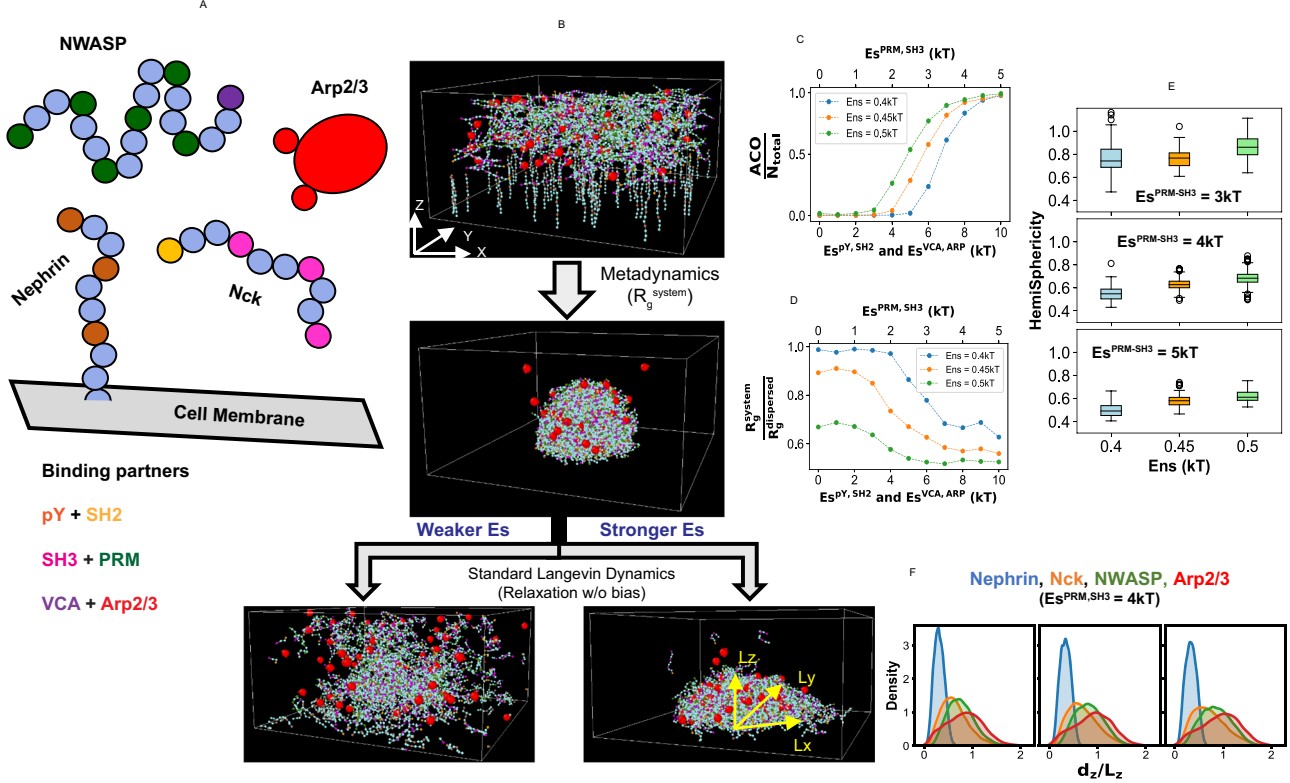

**Fig. 1 | Setup of the Langevin dynamics simulations. A** Coarse-grained bead-spring representations of the molecular components. Some beads serve as binding sites ("stickers") while others constituent the linker regions ("spacers"). The gray surface, containing Nephrin, is the cell membrane. For Nephrin, Nck, and NWASP, light blue beads represent the spacer regions. For Arp2/3, two stickers and the bulky spacer are shown in red. **B** Demonstration of the simulation workflow. Within the simulation geometry, XY surface acts as the membrane which anchors Nephrin. Nck, NWASP, and Arp2/3 can diffuse inside the 3D volume. We initially employed metadynamics simulation (along the order parameter, $R_g^{system}$, as detailed in the method) to facilitate the clustering of molecules to create a hemispherical condensate at the membrane. The condensate is then relaxed using standard Langevin dynamics (without any bias potential) with a series of specific (Es) and non-specific (Ens) interaction energies. The axial lengths (Lx, Ly, and Lz) of a cluster are computed to get its shape. **C** Trend of clustering as a function of Es and Ens. Average cluster

occupancy or ACO measures the mean of the size-weighted cluster size distribution. $N_{total}$ is the molecular count (= 500) present in the system. **D** Trend of radius of gyration ($R_g^{system}$) of the system as a function of Es and Ens. This value is normalized by the $R_g^{dispersed}$ which is obtained by turning off all the attractive interactions (Es = Ens = 0). To obtain the trends in **C** and **D**, 20 snapshots are sampled from a relaxed trajectory. **E** HemiSphericity measures the deviation of the cluster's shape from a perfect hemisphere. For a perfect hemisphere, HemiSphericity = 1. To derive the distributions, for each condition, we run ten stochastic trials and sample ten time-points (10 million steps apart) from each trial; hence each distribution is collected over 100 snapshots (detailed in Method). **F** Spatial location of different molecular types along the cluster radius (details in Method). The z-projection is shown here. Four different colors indicate four molecular types, as indicated in the sub-figure title. Identical to **E**, each distribution is collected over 100 snapshots.

Having the output from these spatial simulations, we built an analytical framework to probe the effect of condensate number on the extent of F-actin production.

Our simulations revealed that, for a fixed amount of Arp2/3, distributing them onto the surfaces of multiple small condensates is a better strategy compared to amassing all the Arp2/3 into a single large droplet. This makes the MCS a desirable solution from the evolutionary perspective. We provide rationale on how similar mechanisms may be operative in other biological condensates to make it a universal function-optimization strategy.

## Results

### Arp2/3 dwells on the surface of hemisphere-like condensates

To probe the structural organization of the condensate, we first consider a simplified bead-spring model of the four-component system (Fig. 1A). Transmembrane receptor protein Nephrin has three phosphotyrosine (pY) sites which interact with the SH2 domain of protein Nck. Nck also contains three SH3 domains which interact with the proline-rich motifs or PRMs that belong to the protein NWASP. The terminal VCA domain of NWASP interacts with one of the two sites on the Arp2/3 complex. These binding domains are interspersed with linker beads to confer flexibility. To mimic the receptor diffusion on a membrane, Nephrin is anchored to the XY plane (Fig. 1B). We start the simulation with a uniform distribution of molecules. Nephrins are distributed on the XY surface (cell membrane), while Nck, NWASP, and Arp2/3 are uniformly placed inside the simulation volume (cytosol).

We first perform a metadynamics simulation to accelerate the molecular clustering process near the membrane (details in Methods). We bias the system along the order parameter, $R_g^{system}$, to create a single cluster with a minimum surface area (Fig. 1B and Supplementary Fig. 1). As a consequence of the multivalent clustering, all the proteins condense near the plasma membrane to create a hemispherical cluster (Fig. 1B). When the order parameter reaches the minimum (hemispherical cluster), we run standard Langevin Dynamics (without bias) to relax the system and quantify the cluster properties.

Next, with the standard (unbiased) Langevin dynamics, we perform a parameter scan to assess the phase diagram of the system. Starting from the hemispherical state, we relax the cluster with a combination of specific sticker-sticker (Es) and non-specific (Ens) energies. As detailed in the method, Es refers to the interaction energies between complementary stickers, while Ens mimics the weak non-specific energy that are present between any pair of particles (stickers and spacers). Below a threshold interaction strength, the cluster dissolves completely (single phase), while above that critical level, the cluster persists but adopts a different shape as a function of Es and Ens (Fig. 1B). While doing an energy parameter scan, Es for SH3 + PRM is maintained as half of Es for pY + SH2 and VCA + ARP (details in method). The clustering state of the system shows a sharp transition as we increase the Es (Fig. 1C), while the system's radius of gyration sharply drops at the same point (Fig. 1D). Comparison of Fig. 1C, D reveals the interplay of Es and Ens in driving the phase transition. At Ens = 0.4kT, the phase transition is majorly driven by the Es, as the system samples the fully dispersed limit at the lower Es (for example, Es$^{PRM-SH3}$ = [1,2,3,4,5 kT]). For Ens = 0.5 kT, even at Es = 0, the system exists at a compact state as reflected in the lower radius of gyration (First green point in Fig. 1D). This combined effect of the energy parameters shapes the phase diagram (Fig. 1C) resulting in different transition points.

Once the cluster is relaxed, we compute its axial lengths (Lx, Ly, and Lz) over multiple snapshots. We then derive a HemiSphericity parameter to measure the cluster's shape. Figure 1E shows that HemiSphericity is close to 1 (Es$^{PRM-SH3}$ = 3kT panel) near the phase transition point. It gradually goes down with stronger Es. For identical Es, stronger Ens yields a higher HemiSphericity.

Next, we determine the relative locations of different molecular types within the cluster. In the simulation volume, X and Y directions are symmetric, but Z is different. Hence, we compute the X, Y, and Z projections of the distance (d) between the molecular centroid and the membrane (details in method). We then normalize the distance components by the respective axial lengths (L) of the cluster. The distance profiles are symmetric along the X and Y directions (Supplementary Fig. 2), but the Z-component is more relevant and informative for our analysis (Fig. 1F). When $d_z/L_z$ > = 1, the molecule is approximately placed near the periphery. Figure 1F shows the overlapping regions of occupancy by different molecular types. Since Nephrin is tethered to the XY plane, it has the lowest $d_z$ (blue distributions). As Nck and NWASP work as adapters between Nephrin and Arp2/3, they occupy the middle part. Arp2/3 is located towards the periphery, as shown by the red profiles. Hence, the Langevin dynamics derived cluster made of Nephrin, Nck, NWASP, and Arp2/3 is hemisphere-like in shape whose HemiSphericity changes with Es and Ens, and Arp2/3 diffuses near the cluster surface.

### Diffusion-mediated local production of F-actin

We next seek to understand the effect of molecular diffusion on actin nucleation. We set up a reaction-diffusion model where Arp2/3 diffuses at the surface of a hemispherical cluster placed on the XY plane. We include the reversible interconversion between GA (G-actin) and FA (F-actin), as shown in Fig. 2A. Actin in polymeric form has ~10 times slower diffusion than the monomeric form[27]. Figure 2B shows a local production of FA at the cost of local depletion of GA. Since Arp2/3 only diffuses in the vicinity of the cluster and FA has a slower diffusion, we observe an accumulation of FA around the cluster. To quantify the local FA concentration near the cluster, we considered a hemispherical shell (Supplementary Fig. 3) where FA accumulation takes place. We see a significant difference between the bulk and the surface concentration of FA (Fig. 2C). It is to be noted that the bulk concentration incorporates all the volume elements, including the cluster surface. Interestingly, the surface enrichment of FA depends on the diffusion of FA itself (Fig. 2C). When we double the FA diffusion coefficient, the concentration difference of FA between surface and bulk goes down (Supplementary Fig. 4). To probe this trend further, we gradually decrease the FA diffusion coefficient, keeping GA diffusion constant. Figure 2D reveals a pattern of FA surface enrichment as a function of differential diffusion of FA. With slower FA diffusion, local accumulation of FA becomes more prominent. This FA production varies when we consider hemispheroidal clusters with varying degree of HemiSphericity (Fig. 2E). We can adjust the radius and height of the perfect hemisphere to create an object with lower HemiSphericity and identical volume (Supplementary Fig. 5). These hemispheroidal clusters have larger surface area; consequently, surface density of Arp2/3 gradually goes down with lower HemiSphericity, if we keep the total Arp2/3 count fixed. This lowering of Arp2/3 density is reflected in the gradual decrease in FA production. Figure 1E informed us that, within the wider parameter range explored, the lower bound of HemiSphericity is around 0.4. The reaction-diffusion model predicts that the surface enrichment effect is present even with a cluster of 0.4 HemiSphericity. Hence, Arp2/3 mediated local production of FA at the cluster surface is a kinetic phenomenon that stems from the slower diffusion of polymeric actin compared to its monomeric counterpart. The shape of the cluster affects the magnitude of local enrichment, but the qualitative behavior remains the same.

### Multi-cluster state yields greater F-actin production

We next use our reaction-diffusion model to compare a multi-cluster state (MCS) with its single-cluster counterpart. Figure 3A illustrates the simulation setup where we have four hemispherical clusters placed on the XY plane. The radius of each cluster is adjusted such that the total volume occupied by four clusters is the same as the single large cluster

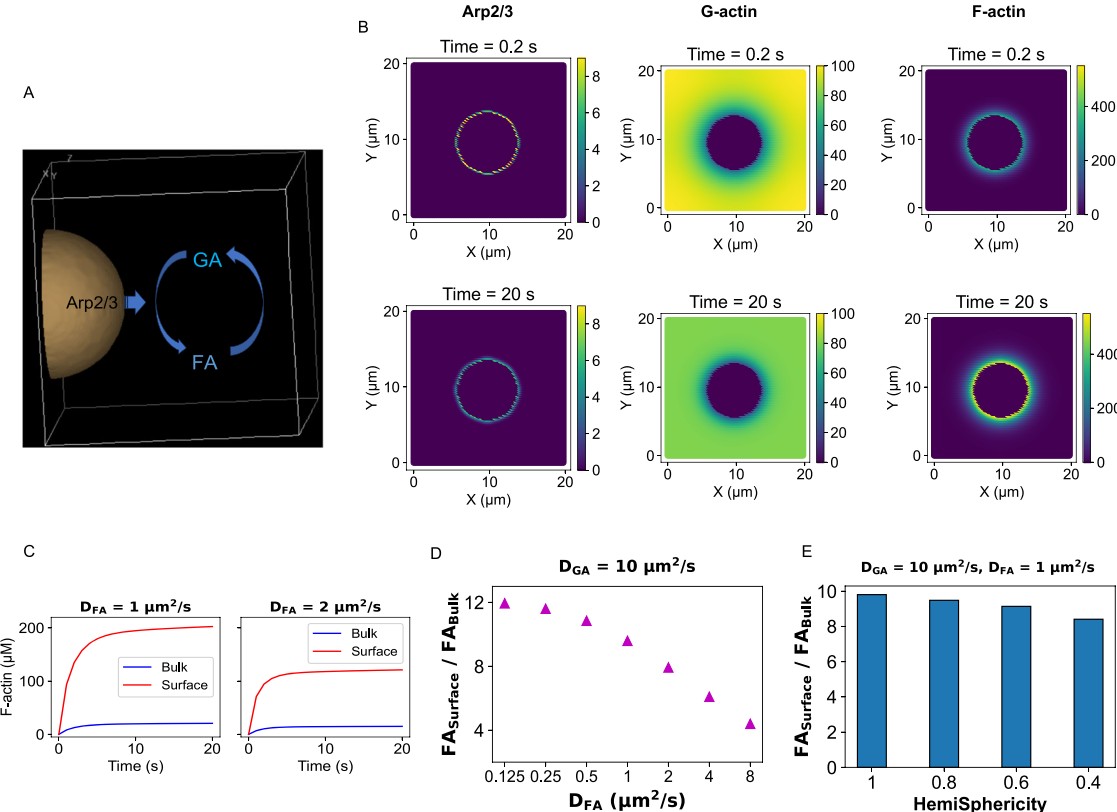

**Fig. 2 | Reaction-diffusion model with a single condensate. A** Illustration of the simulation geometry. The "Virtual Cell" software is used to define the geometry and set up the simulation conditions. The hemispherical condensate is situated on the XY plane. Arp2/3 only diffuses around the condensate surface. Actins (both G and F) can diffuse in the entire region. **B** Concentration profiles (shown as color bars) of the model components at the beginning of simulations (Time = 0.2 s, upper panel) and the steady state (Time = 20 s, lower panel). These snapshots represent the plane at Z = 0. Diffusion constants, $D_{GA}$ = 10 μm²/s and $D_{FA}$ = 1 μm²/s. **C** Timecourse of F-actin (FA) concentration at two FA diffusion constants ($D_{FA}$). To compute the "Surface concentration", a hemispherical region of interest (larger than the condensate) is considered (details in method) and the amount of FA in the shell volume measures the surface concentration. The "Bulk concentration" incorporates all the volume elements, including the condensate surface. **D** Measure of local production of F-actin as a function of differential diffusion of actin in monomeric ($D_{GA}$) and polymeric ($D_{FA}$) form. We keep $D_{GA}$ (10 μm²/s) fixed and vary $D_{FA}$. **E** Trend of local FA production as a function of the condensate's HemiSphericity.

considered in Fig. 2A. The number of Arp2/3 on each cluster is also divided by 4 such that the total Arp2/3 count is the same as Fig. 2A. So, we have a comparison between one-cluster and four-cluster scenarios where total volume of clusters and total Arp2/3 count are the same. The question is, which configuration is more efficient in FA production?

Figure 3B shows that FA production takes place near the surface of each cluster. Again, due to Arp2/3 localization in the cluster and slower diffusion of FA, we observe a local FA accumulation around the clusters. To compute the FA concentration near the cluster surfaces, we again consider a hemispherical shell around each cluster. The shell volumes are adjusted in such a way that the total shell volume for the four-cluster state is roughly the same as the single-cluster state. When we quantify the FA concentrations, we notice an interesting trend (Fig. 3C). For the four-cluster configuration, local FA production (FA per cluster) is slightly lower (red line) than the single cluster analog (black dotted line, derived from Fig. 2C). Since there are four clusters present in the system, the global output = 4 * local output. The blue line (Fig. 3C) depicts that the steady state level of global FA, for the four-cluster state, is greater. This behavior is again dependent on the FA diffusion constant (two panels, Fig. 3C), which is consistent with the notion that the local FA production is a kinetic phenomenon. If we systematically probe the relative gain in FA production, Fig. 3D shows that the global gain in a four-cluster state gradually goes down with faster FA diffusion. It is to be noted that the configuration with $D_{FA}$ = 1 μm²/s and $D_{GA}$ = 10 μm²/s is closest to the physiological condition. The gain in the four-cluster state remains similar for clusters with

lower HemiSphericity (Fig. 3E). Hence, slower diffusion-mediated local production of FA makes the four-cluster state more efficient compared to the single large cluster.

## Systems theoretic analysis reveals an optimal cluster count
The reaction-diffusion model informs us that (a) FA gets produced locally near the cluster and (b) MCS produces greater FA than its equivalent single cluster analog. We then seek to probe this trend further with a systems theoretic approach. We first express the n-cluster state by the following ordinary differential equation (ODE):

$$\frac{dF_{local}}{dt} = k1 * G_{local}^2 * \rho(n) * F_{local} - k2 * F_{local} \tag{1}$$

Where

$$G_{local} + F_{local} = C(constant) \tag{2}$$

The GA to FA conversion happens locally near the cluster with the constraint of mass conservation (Eq. 2), $G_{local} + F_{local} = Constant$. $G_{local}, F_{local}$ are local concentrations (near the cluster surface) of actin in monomeric and polymeric form, respectively. $\rho(n)$ is the surface density of Arp2/3 on each of the n clusters. k1 and k2 are rate constants for polymerization and depolymerization. Branching of F-actin requires a pre-existing filament (so-called "mother filament"), one Arp2/3 complex, and a couple of G-actin. Hence, the branching rate is expressed by the mass-action kinetics described by Eq. (1). We note that F-actin can

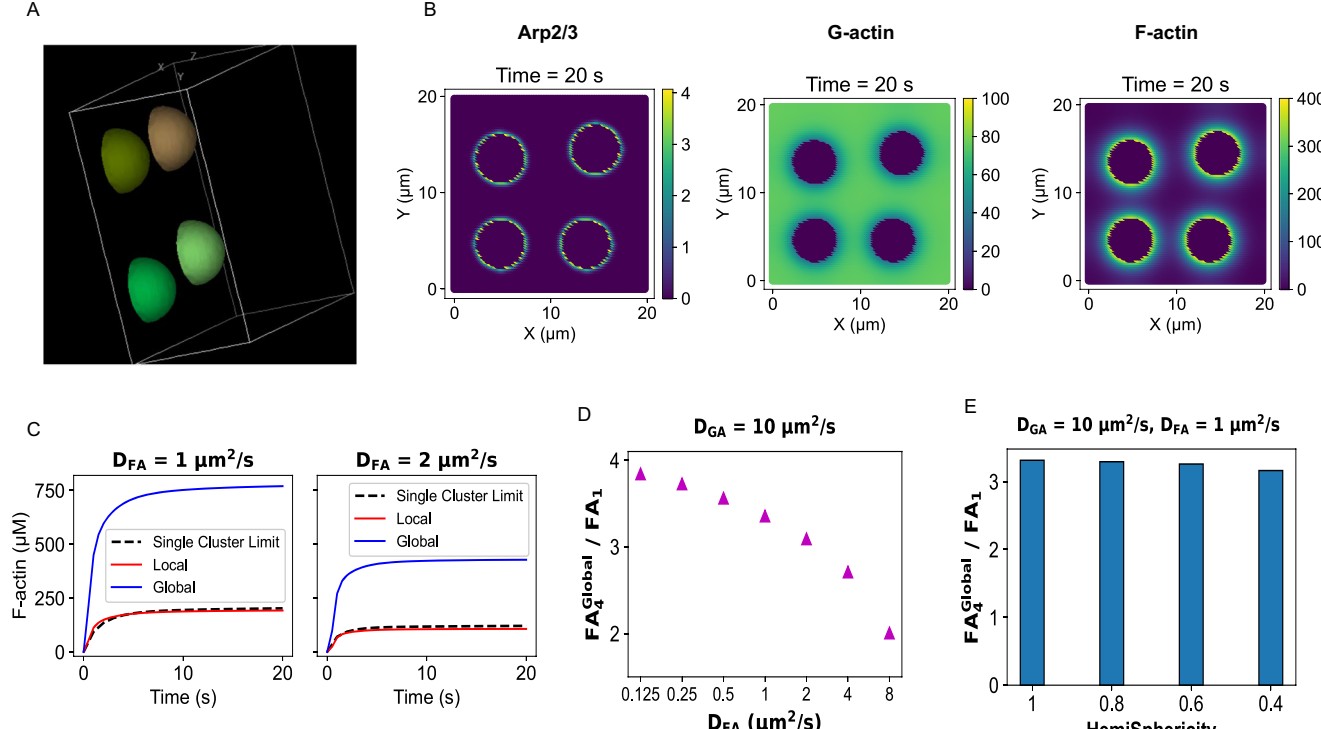

**Fig. 3 | Reaction-diffusion model with multiple condensates. A** Illustration of the simulation setup. This snapshot from the "Virtual Cell" displays four hemispherical condensates situated at the XY plane. Similar to Fig. 2A, Arp2/3 only diffuses around the condensate surface. Actins (both G and F) can diffuse in the entire region. **B** Concentration profiles (shown as color bars) of the model components at the steady state (Time = 20 s, last timepoint). These snapshots represent the plane at Z = 0. $D_{GA} = 10\ \mu m^2/s$ and $D_{FA} = 1\ \mu m^2/s$. **C** Timecourse of F-actin (FA) concentration at two FA diffusion constants ($D_{FA}$). "Local" refers to the F-actin (FA) produced at each condensate surface, while "Global" is the cumulative FA produced at four-condensate surfaces. To compute the surface concentration, as in Fig. 2, a hemispherical region of interest (larger than the condensate) is considered (detailed in Method). The black dotted line ("single cluster limit") indicates the FA concentration (surface FA from Fig. 2C) when an equivalent amount of Arp2/3 is concentrated on the surface of a single large condensate. **D** Quantification of relative gain (in FA production) as a function of differential diffusion of actin in monomeric ($D_{GA}$) and polymeric ($D_{FA}$) form. $FA_4^{Global}$ indicates the global FA concentration (blue lines in C) in the four-cluster state, while $FA_1$ is the single-cluster limit. **E** Quantification of FA production gain in a four-condensate state as a function of individual condensate's HemiSphericity.

autocatalyze its own production. From Eqs. (1) and (2), we also note that the overall rate depends on $G_{local}$ in a nonlinear manner.

At the beginning, we assume each cluster to be hemispherical such that (Supplementary Text 1)

$$\rho(n) = \alpha^* n^{-1/3} \qquad (3)$$

where α is the surface density of Arp2/3 when $n = 1$.

We first analyze the single cluster case ($n = 1$). Figure 4A displays the roots of the right-hand side of Eq. 1 and their stability as a function of α. The upper two panels indicate a stable solution of F = 0, while for the lower two panels, the stable roots vary as we change α. Figure 4B clearly shows that there is a critical value of α, beyond which the system converges to different non-zero F-actin levels. In other words, the surface density of Arp2/3 serves as a tunable bifurcation parameter that switches the system from a non-responsive ($F = 0$) state to a responsive ($F > 0$) state. Stability analysis of the roots of the r.h.s. of Eq. 1 yields (Supplementary Text 2)

$$\alpha_{critical} = \frac{k2}{k1^* C^2} \qquad (4)$$

For the parameters ($k1 = 0.001$, $k2 = 1$, and $C = 100$) used in the model, $\alpha_{critical} = 0.1$, consistent with the bifurcation behavior in Fig. 4A, B. For a given dissociation rate parameter ($k2 = 1$), Fig. 4C shows the state space of $\alpha_{critical}$ as a function of polymerization rate

(k1) and the total amount of actin pool (C). For a fixed value of k1, if we increase C (move along the column), the value of $\alpha_{critical}$ goes down; that is, even lower density of Arp2/3 can trigger polymerization in the presence of a higher amount of actin pool. Similarly, for a given amount of C, higher k1 (move along the row) causes $\alpha_{critical}$ to go down. Overall, Fig. 4A–C show that, to initiate the actin-branching, each cluster needs to recruit a threshold amount of Arp2/3 such that the surface density exceeds $\alpha_{critical}$. The extent of $\alpha_{critical}$ depends on the reaction rates and amount of total actin pool available near the cluster.

Now we consider a hemispheroidal cluster which has Hemi-Sphericity, $hS < 1$. Supplementary Fig. 6 showed how to create hemispheroidal clusters with a shape parameter $f$, where $0 < f < 1$. For a perfect hemisphere, $f = 1, hS = 1$. With lowering the value of $f$, one can create clusters with arbitrary HemiSphericity, such that, $hS = f^{3/2}$.

For a fixed number of Arp2/3, due to higher surface-to-volume ratio (Supplementary Fig. 7), surface density of Arp2/3 goes down for lower hS (Supplementary Text 3). Figure 4D reflects this trend where the same number of Arp2/3 is recruited onto the clusters with varying hS.

We next want to investigate the causal factor underlying the greater response coming from the MCS. Figure 3 informed us that if we split a large cluster into four smaller clusters and distribute N Arp2/3 equally into each cluster (such that each smaller cluster retains N/4), the four-cluster scenario produces a higher F-actin response compared to the single large cluster. Although the volumes of clusters are the same in both cases, the surface-to-volume ratio increases as we split one hemisphere into n smaller ones. To probe this effect further, we

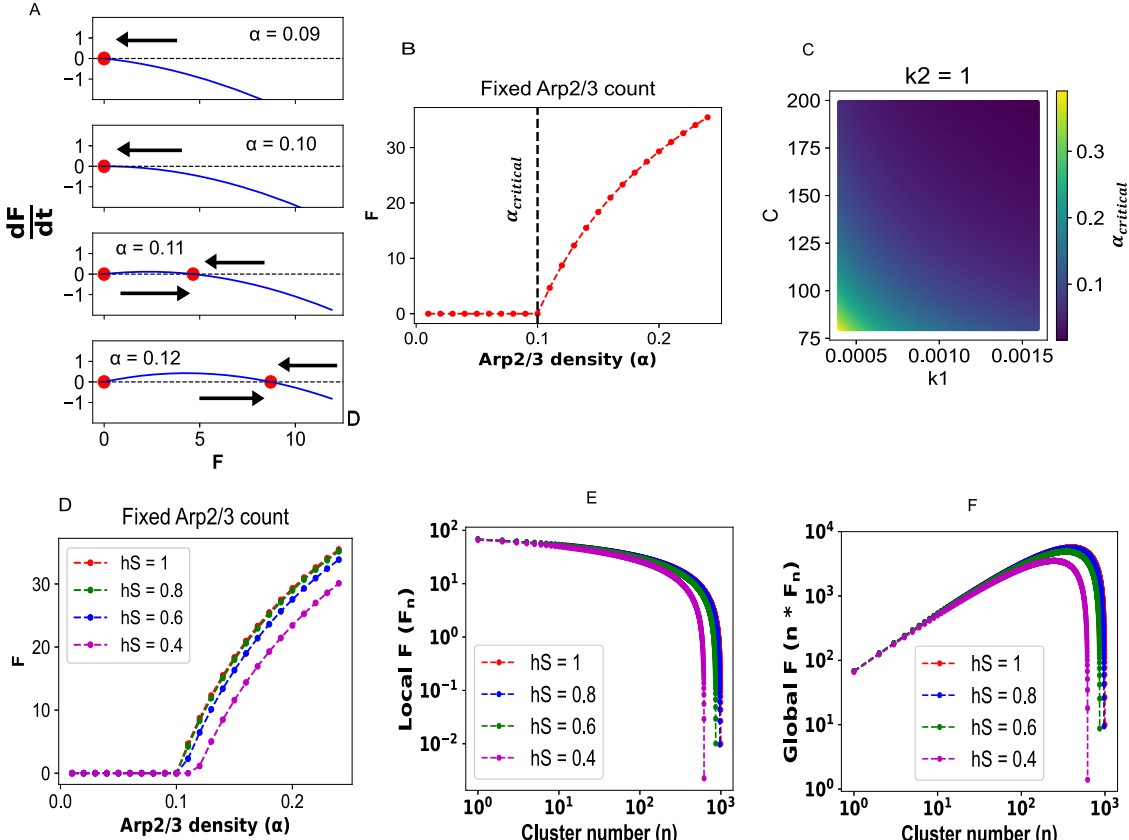

**Fig. 4 | Analytical solution of the ODE model.** For **A–C**, a perfect hemisphere is considered. **A** Bifurcation analysis. Each panel shows the functional value of $\frac{dF}{dt}$ (Eq. (1)) at a given value of Arp density (α). The red circles refer to the "roots" of Eq. (1), that is, the values of F where $\frac{dF}{dt} = 0$. The arrow represents the direction of "flow"; it is rightward for a region where $\frac{dF}{dt} > 0$ and leftward otherwise. Flow towards a root signifies stability. **B** Solution of F-actin (F) as a function of Arp density. The dotted line indicates the bifurcation point where F production starts to happen. The density of Arp at this point is termed $\alpha_{critical}$. **C** Value of $\alpha_{critical}$ (color-coded) as a function of k1 and C (Eq. (2)). **D** Trend of Arp-dependent F production as a function of the cluster's HemiSphericity. Creating a spheroid-like geometry from a sphere increases the surface-to-volume ratio. For a fixed number of Arp, the surface density of Arp goes down with smaller HemiSphericity (Supplementary Text 3). **E** Amount of F produced per cluster in a n-cluster system. **F** Amount of cumulative F, which integrates contributions from all the clusters. Different colors represent distinct HemiSphericity.

first derive an expression for local F production (F per cluster in the n-cluster system) (Supplementary Text 4),

$$F_{local} = C - \sqrt{\frac{1}{\alpha}\left(\frac{k2}{k1}\right)} * K(f)^{1/2} * n^{1/6} \quad (5)$$

Where

$$K(f) = \frac{1}{2}\left(\frac{1}{f} + \frac{f^2}{2e}\log\left(\frac{1+e}{1-e}\right)\right) \quad (6)$$

Where

$$e = \sqrt{(1 - f^3)} \quad (7)$$

The shape parameter, $f$ decides the HemiSphericity of the cluster. For a perfect hemisphere, $f = 1, K(f) = 1$. Using Eq. (5), we compute the local F as a function of cluster number ($n$). Figure 4E shows a rather interesting trend. As we increase n, initially the drop in per-cluster F is minimal. Consequently, on the cumulative global scale (Fig. 4F), we have a gradual increase in F production. We note the logarithmic scale along both the axes. As we go to large cluster numbers, beyond a certain value, local F starts to drop quickly which also drags down global production. The differential rate at which local F drops

as a function of $n$ yields the most fascinating aspect of the global F trend − an optimal cluster number ($n_{optimal}$). Figure 4F suggests that there exists a multi-cluster configuration where the F-actin production, on the global or system level, is maximum. This behavior is consistent with a wide range of hS. Although, with lower hS, the maximal level of global F gradually drops and $n_{optimal}$ shifts towards lower values (Fig. 4F).

We notice that for constructing the scaling laws (Supplementary Texts 3, 4), we used an oblate spheroid geometry which does not preserve a constant curvature. In an alternative approach (Supplementary Text 5), one can deform the cluster shape maintaining a spherical surface. This constant curvature construction produces a qualitatively similar relationship between the surface areas of a perfect hemisphere and a spherical cap. We also note that while computing global F, we assumed all the clusters to have identical shapes, which is characterized by the $K(f)$ parameter. In other words, for hS = 0.6, a large and a small cluster will have a hemisphericity of 0.6. It is a reasonable approximation given the fact that the shape of a condensate should be determined by the binding energies between the membrane-bound and cytosolic molecules, which remains independent of the condensate size.

Upon close inspection of Eq. (1), we realize that the minimal drop in local F (Fig. 4E) stems from the highly nonlinear nature of the F production. Figure 4B shows that each cluster needs to have a threshold Arp2/3 density ($\alpha_{critical}$) to "turn on" the F production switch.

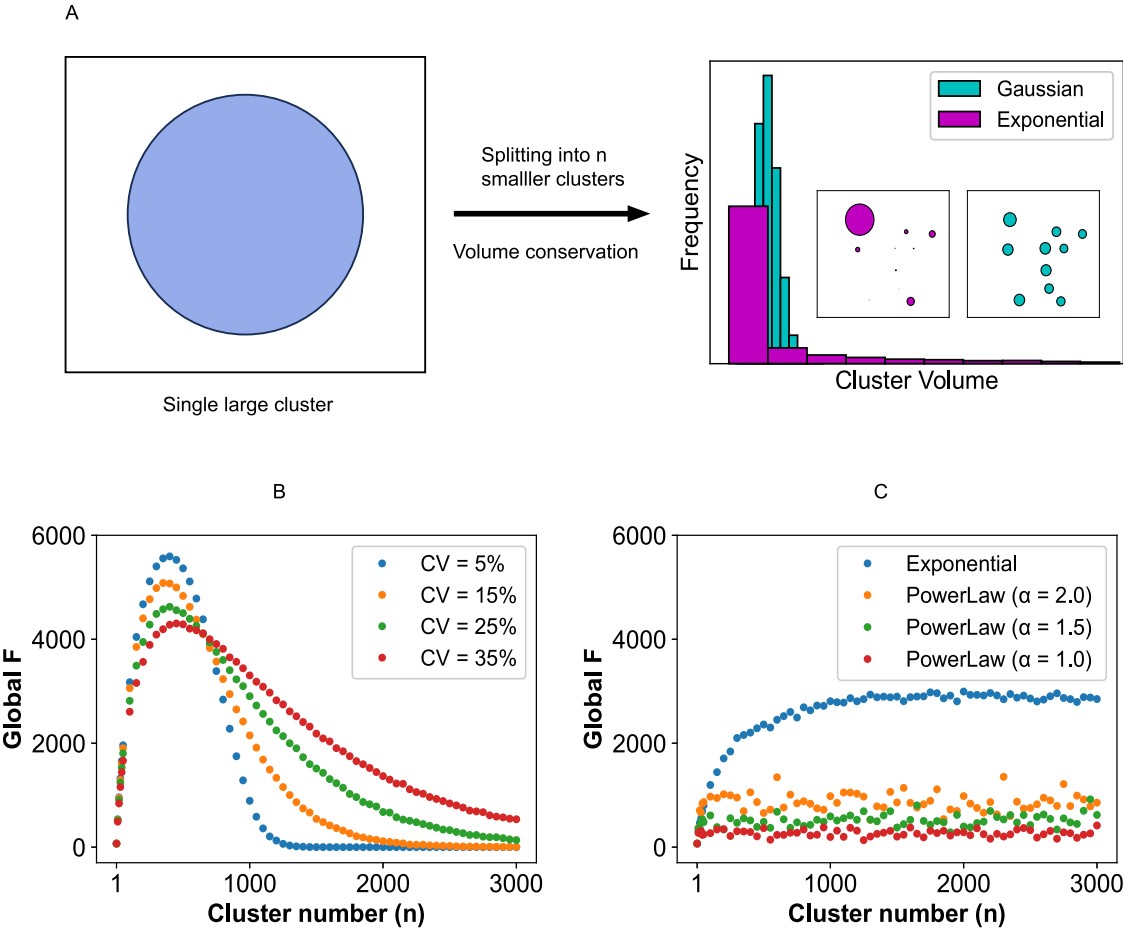

**Fig. 5 | Effect of cluster size distributions on global F-actin production.**
**A** Illustration of choosing the distribution of cluster sizes. When we split one large cluster (radius = R) into *n* clusters of equal size, the radius of each small cluster, $r_{small} = n^{-0.33} * R$. We create a distribution (mean = $r_{small}$) of cluster radii in a such way that the total volume is conserved during 1-to-n splitting process. On the right side, gaussian and exponential distributions are shown. Inset displays the single frames drawn from each of the two distributions (*n* = 10). **B** When cluster size follows a Gaussian distribution, a trend of global F-actin production at four different coefficients of variation (CVs). $CV = (\sigma/\mu)*100$, where $\sigma$ and $\mu$ are the standard deviation and mean of the Gaussian distribution, respectively. CV measures the width of the distribution. $\mu = r_{small}$ for all cases. **C** Trend of global F-actin productions when cluster sizes conform to an exponential or a power-law distribution. To create an exponential distribution, we need a rate parameter ($\lambda$), such that, $f(x,\lambda) = \lambda * \exp(-\lambda * x)$. We set the mean, $1/\lambda = r_{small}$. To create a power-law distribution, we need a scale (m) and shape ($\alpha$) parameter, such that, $f(x,\alpha,m) = \frac{\alpha * m^{\alpha}}{x^{\alpha+1}}$. we set m = $r_{small}$, and vary $\alpha$ (2, 1.5, 1). To compute F-actin production, we generate ten distributions for each *n*.

Beyond that point, due to the inherent nonlinearity in the production rate, putting more Arp2/3 onto the same cluster is less efficient; rather, creating a second reaction center (cluster) with $\alpha_{critical}$ Arp2/3 is a more prudent strategy. Recursively, if we follow the same scheme, creating more clusters produces more F-actin. Since we have a finite amount of Arp2/3 to distribute amongst n clusters, beyond a parameter-specific value of *n*, Arp2/3 density on each cluster surface falls below the critical level and F production gets "switched off". Hence, the existence of a threshold Arp2/3 density and nonlinear production of F-actin makes the multi-cluster configuration more efficient. Moreover, irrespective of the cluster shape, an optimal number of clusters seems to be the best functional solution.

### Statistical distribution of cluster sizes yields distinct F-actin production profiles

In the previous section, all the clusters are assumed to be of the same size. In reality, we expect a size distribution. So, we next ask what happens to the F-actin profile if we consider a size distribution of hemispherical clusters? We generate distributions with the constraint of volume conservation (Fig. 5A). The recruitment of Arp2/3 to a cluster is proportional to its volume (detailed in the Method section).

We first consider a Gaussian distribution. For an n equal-sized-cluster state, if $r_{small}$ is the radius of each cluster, the distribution is created with the mean of $r_{small}$ and a prescribed coefficient of variation (CV). Figure 5B shows the trend of global F production at four different CVs. In all four cases, the optimal cluster count, where global F reaches maximum, remains similar. However, the shape of the global F curve becomes more asymmetric as we make the size distribution wider (higher CV). Supplementary Fig. S8A, B reveals that the wider Gaussian distribution, even at larger n, can sample a few points above the bifurcation threshold that produce non-zero F-actin. This results in a longer tail in the global F-actin production (Fig. 5B). From a biological perspective, a wider distribution provides a wider "responsive range", that is, extent of global F is held at a non-zero level for a wider range of cluster sizes.

Lee et al.[28] recently quantified the size distributions of natural and synthetic condensates in living cells. They reported that the physiological condensates conform to an exponential distribution, while pathological aggregates follow a power-law-like distribution. So, we wanted to find out our model predictions if we prescribed exponential or power-law distribution for the cluster size. Figure 5C suggests that an exponential distribution produces a distinct F-actin profile which grows initially and beyond a cluster count, it plateaus.

If we compare this with the outcome of Gaussian distribution, the maximal level of global F is roughly half. However, global F is retained at the maximal level for a broad cluster size range. For example, $n = 1000$ produces ~3000 times more F than at $n = 1$. For a power-law distribution, the system simply fails to produce a significant response. With a larger exponent ($\alpha$), which signifies smaller tails, the response improves. Supplementary Fig. S8C, D show that the long-tail nature of both exponential and power-law distributions allows sampling of fewer points above the bifurcation threshold even at larger n. However, purely due to the statistical sampling, an exponential distribution can produce higher global F, while the power-law distribution with an elongated tail exhibits a significantly lower level of global F.

## Discussion

The existence of multiple biological condensates directly contradicts the prediction of polymer physics that states that, at equilibrium, one large droplet should coexist with the soluble phase. Since biological systems operate directly under evolutionary pressure, we investigate the following problem—given a fixed number of molecular components, how to distribute them into spatial clusters to achieve maximal output? The output could be system-specific or context-dependent. But, we seek to uncover a generic principle that may incentivize the stabilization of a multi-droplet state. Using a combination of three complementary computational techniques, we have established that, indeed, multiple droplets are functionally more efficient compared to their single droplet counterpart.

We have used the condensation of signaling proteins (Nephrin, Nck, NWASP, and Arp2/3) upstream to actin nucleation pathway as a case study. Langevin dynamics revealed the emergence of hemispherical condensate whose shape is determined by the specific and non-specific interaction energies. It also showed that Arp2/3 resides on the condensate surface (Fig. 1). We then used a hemispherical solid geometry as a proxy for the condensate where Arp2/3 diffuses on the surface. The reaction-diffusion model predicted a local production of F-actin which depends on the differential diffusion of actin in G and F form (Fig. 2). This local enrichment of F-actin makes a multi-cluster configuration more efficient in producing F-actin (Fig. 3). The diffusion-mediated localized F-actin production enabled us to write an ordinary differential equation (Eq. 1) where G-actin locally gets converted into F-actin in an Arp2/3 dependent manner. Our theoretical analysis revealed that the surface density of Arp2/3 serves as a functional switch that triggers F-actin production beyond a threshold density (Fig. 4). Due to the highly nonlinear production rate, having many clusters, each equipped with a threshold amount of Arp2/3, creates an optimal scenario where F-actin production is maximum. Depending on the cluster size distributions, the F-actin production profile, on the system scale, takes distinct shapes (Fig. 5) that may be related to the disease phenotype of cellular systems. This study provides an example of how multiple computational tools can synergistically be employed to gain a holistic understanding of a biological phenomenon. The ODE model is informed by the reaction-diffusion simulations which are again motivated by the Langevin dynamics output. Thus, it exemplifies how to amalgamate scale-specific insights into a multi-scale picture.

The shape of the condensate emerges as an important parameter in our study. Figure 1E informed us that the HemiSphericity of the membrane-bound condensate can vary between 0.9 to 0.4 based on the values of specific and non-specific energies. Both the reaction-diffusion model (Figs. 2 and 3) and ODE analysis (Fig. 4) suggested that a higher HemiSphericity yields greater response compared to the oblate hemispheroids due to the smaller surface-to-volume ratio (Supplementary Fig. 7). It is interesting to notice that the HemiSphericity of condensate is close to 1 right around the phase transition boundary (Fig. 1B, E). One can speculate that the interaction energies

between the binding sites may have evolved to be a tunable handle to maximize the condensate HemiSphericity.

The organization of binding sites within a protein sequence determines the mesoscopic architecture of the condensate. In our system, the spatial arrangement of SH2, SH3, PRM, and VCA dictates the peripheral location of Arp2/3 which is optimal for the desired downstream effect (F-actin production). If we shuffle the arrangements of binding domains/motifs, the condensate will likely assume a different spatial architecture. It is intriguing to think that for a set of condensate-forming proteins that participate in a biochemical task (signaling pathway, for example), the sequences may have co-evolved to yield a condensate structure that can facilitate downstream processes. Given the spatial heterogeneity of almost all known biomolecular condensates (P granules, stress granules, nucleolus, etc.), our functional perspective provides a systematic way to think about the core and the interface of these condensates. It might also allow us to rationalize the relative location of the complementary binding sites in the partner molecules that form a condensate.

We have previously proposed that the multi-cluster state or MCS is a kinetically arrested metastable state[7]. With recent experimental validations[29,30], it is becoming evident that spatially heterogeneous multi-phasic condensates, as well as the long lifetime of multiple condensates, have a strong kinetic component. In the current study, we have identified that the downstream reaction module, like F-actin production, is also controlled by kinetic effects like diffusion. This coupling of kinetic effects in upstream and downstream layers makes multiple condensates an optimum solution. Given the necessity of spatiotemporal regulation of cellular biochemistry, it is not surprising that living cells have evolved to take advantage of the kinetics or timing of events.

Our analysis underscores the functional importance of the size distribution of clusters. The system considered here is responsible for cell signaling. For instance, the rate of actin branching at the leading edge of a moving cell dictates the speed and efficacy of cell migration. Different cluster size distributions essentially produce distinct response patterns (Fig. 5). By creating a more asymmetric distribution (from Gaussian to exponential), a "spiked" response becomes more "sustained". That is, the responsive regimes (cluster number) of FA production get broader, at the expense of reduced maxima, with increasing variance of cluster sizes. However, there is a limit to this process—creating a fat-tailed distribution like a power-law might trigger a pathological state which simply cannot respond to the external signal. This is consistent with recent experimental findings[28]. Depending on the spatiotemporal context, different cellular systems can exploit these distributions to achieve specific tasks. It is noteworthy that several physical mechanisms like active processes[31,32], Pickering agent-like surface adsorption[18], growth-dependent slower diffusion[33], etc. have been proposed to explain the condensate size distribution. One or multiple of these mechanisms may be operative in stabilizing the multi-condensate state which can be advantageous from a functional point of view.

The notion of spatial clustering of biomolecules to optimize the function has been proposed in other biochemical contexts. Co-clustering of multiple metabolic enzymes[34] provides a strategy to efficiently channel the metabolic intermediates. Interestingly, there is an optimal cluster size where metabolite processing is maximum. A recent study[35] probed the effect of condensate size on enzyme-catalyzed reaction rates. By titrating the concentration of enzyme-conjugated IDRs, the authors created two systems, one with nano-condensates (just above saturation concentration) and the other with micro-condensates (way above saturation concentration). They reported that the enzymatic rate enhancement is higher in nano-condensates compared to the micro-condensates. This observation is well aligned with our theoretical predictions. Although our system of investigation involves a reaction that takes place at the condensate

surface and conforms to a mass-action type kinetics, the general notion of multiple smaller clusters being functionally more efficient than fewer larger clusters seems relevant. Since non-linearity and feedback loops are quite common in biochemical networks, we believe that our proposed framework will help rationalize observations in a wide range of biochemical systems.

In summary, our multi-scale analysis offers a compelling justification for the presence of multiple condensates within living cells. For a function-optimization incentive, the sequence of the individual molecules, as well as the structure of the molecular network, may have evolved to take advantage of the kinetic effects in maximizing the desired output.

## Methods

### Langevin dynamics

**Bead-spring representation.** We used bead-spring polymers for the coarse-grained representation of the signaling proteins (Fig. 1A). Nephrin, Nck, and NWASP all contain intrinsically disordered regions (IDRs) in their sequences[36], but Arp2/3 complex is a large structured protein made with seven subunits[37]. Each component is modeled with a minimal description that includes a known number of stickers (valence) and, except for Arp2/3, a pair of spacers between the stickers to mimic the flexibility imparted by the IDRs. For Arp2/3, we have two stickers connected by a bulky spacer to model the steric effects coming from the large seven-subunit protein complex.

**Force field.** Each polymer contains the prescribed number of beads connected by harmonic bonds. Stretching energy of each harmonic bond, $E_{bond} = K_b * (R - R_0)^2$ where $K_b$ is the spring constant and $R_0$ is the equilibrium bond distance. R measures the distance between the bonded beads at any given time. Each bond is a harmonic distance constraint that enables permanent connectivity within a chain. In our model, $R_0 = 10$ Å and $K_b = 3 \frac{kcal}{mol*Å^2}$.

To allow chain flexibility, the angle (θ) between three successive beads is modeled with a cosine function: $E_{bending} = \kappa * (1 - \cos\theta)$ where κ decides the bending stiffness. In our model, $\kappa = 1 kcal.mol^{-1}$.

Each pair of beads interacts via a non-bonded isotropic interaction, modeled by Lennard-Jones (LJ) potential: $E_{LJ} = 4 * \epsilon * [(\frac{\sigma}{r})^{12} - (\frac{\sigma}{r})^6]$ where σ represents the bead diameter and r is the separation between the beads. ϵ is the depth of energy-well that determines the strength of attractive potential. To distinguish from the single-valence specific interactions (described below), we refer to ϵ as Ens or "non-specific" interaction energy (Supplementary Fig. 9). To achieve computational efficiency, the LJ potential is neglected beyond a cut-off distance ($R_{max}$). In our model, except for the Arp2/3 spacer, σ = 10 Å. For the bulky Arp2/3 spacer, σ = 30 Å. For all pairwise LJ interactions, $R_{max} = 2.5 * \sigma$.

**Modeling specific interactions.** To encode "specific" interactions between complementary sticker types, we introduced inter-sticker bonds (Supplementary Fig. 9). When two stickers approach within a cut-off radius ($R_{cut}$), they can form a "bond" with a probability, $p_{on}$. The bond can be broken with a probability of $p_{off}$, at or greater the distance of $R_{cut}$. These are "specific" interactions because once a sticker pair is bonded, they can't form another bond with complementary stickers that are still within $R_{cut}$. In other words, each sticker has a valency of 1. The bonds at distances shorter than $R_{cut}$ are modeled with a shifted harmonic potential:

$$E = \frac{E_{min}}{(R_0 - R_{cut})^2} \left[ (R - R_0)^2 - (R_{cut} - R_0)^2 \right]$$

R is the inter-sticker distance. At the resting bond distance ($R_0$), the energy is $-E_{min}$. We refer to this parameter as Es or "specific" energy. When two complementary stickers form a bond, the gain in energy is Es. In other words, the depth of energy potential is Es at the resting distance. As defined in the previous section, Lennard-Jones potential is present between any pair of beads, including stickers. However, LJ is ignored if two beads are connected via bonds. With this scheme, when two stickers form a bond, LJ interactions between them are turned off. Hence, for a bonded sticker pair, Ens is overridden by Es. We also note that, at $R = R_{cut}, E = 0$. For $R > R_{cut}, E$ is set to be zero. In our model, $R_0 = 1.122 * \sigma$, $R_{cut} = R_0 + 1.5$ Å, $p_{on} = 1$, $p_{off} = 1$.

Since both the probabilities ($p_{on}$, $p_{off}$) are set to 1, the stochastic factor of inter-sticker binding and unbinding is absent. For a probability <1, there is a stochastic factor that determines whether to make or break the bond even when the distance criteria is satisfied. In our case, the bond formation or breakage only depends on the inter-sticker distance. The lifetime of the bond becomes a function of Es, such that, $\tau_{bond} \propto e^{Es/kT}$. Indeed, the inter-sticker dissociation events decay exponentially with higher Es (Supplementary Fig. 10), consistent with an Arrhenius-like rate expression, $Rate \propto e^{-Es/kT}$. This also indicates that the lifetime of individual bonds is sufficient to ensure thermalization within the harmonic well (whose depth is Es) such that detailed balance is obeyed. Since the inter-sticker association rate is a number determined by the particle diffusions and dissociation is a process that requires overcoming the energy barrier of the sticker-sticker bond well, the Arrhenius rates (Supplementary Fig. 10) are indicative that stickers are thermalized in their corresponding wells.

We have three types of binding interactions: (1) pY and SH2, (2) SH3 and PRM, (3) VCA and Arp2/3. The binding affinity between phosphotyrosine (pY) and SH2 may vary in the sub-micromolar range; however, a typical value is reported to be ~1 μM[38]. Interactions between VCA and Arp2/3 sites also have an affinity of ~1 μM[39]. The binding strength of SH3 and PRM is relatively lower and can vary between 1 to 200 μM[40]. A typical value can be considered around 100 μM. To account for the relative interaction strengths, we have used two values of Es in our model. Since, $\Delta G_{binding} \propto \log(Kd)$, we have set the Es for SH3 + PRM two times lower than that of pY + SH2 and VCA + Arp2/3 (Fig. 1C, D). While titrating Es to create a phase diagram, we maintain the difference of a factor of 2 between $Es^{PRM,SH3}$ and $Es^{pY,SH2}$ (or $Es^{VCA,ARP}$) to mimic the 100 times difference in binding affinities.

For energy parameters (Es and Ens), we report the values in the unit of $k_BT$ where $k_B$ is the Boltzmann constant, and T is the system's temperature. We use "kT", in place of "$k_BT$", for notational simplicity. We note that, 1 kT ~ 0.6 kcal/mol.

**Modeling confined diffusion.** Since Nephrin is a transmembrane receptor protein, our model Nephrin contains an "anchor" bead which is tethered on the XY surface such that it can only diffuse in two-dimensional plane. This is achieved by setting the z-component of force and velocity (applied on the anchor bead) to zero.

**Simulation details.** We used the LAMMPS package[41,42] to execute Langevin Dynamics simulations. We performed our simulations in a rectangular cuboid box (X = 600 Å, Y = 600 Å, Z = 300 Å). XY face (and the opposite face by symmetry) has a reflective boundary condition to model the steric hindrance from the membrane, that is, when a particle hits the membrane, it gets bounced off. The other four faces (XZ and YZ) of the box have periodic boundary conditions. The simulation temperature is 310 K. The viscosity of the simulation medium is described with a "damp" parameter which is set to 500 femtoseconds (fs). For the membrane-bound anchor bead, damp = 50 fs since membrane diffusion is slower than cytosolic diffusion. The damp parameter is inversely proportional to the viscosity of the solvent. Simulation timestep = 30 fs. We start our simulation with a uniform molecular distribution: 80 Nephrins on the membrane (XY plane), and 240 Nck + 120 NWASP + 60 Arp2/3 in the simulation volume.

We used metadynamics simulations[43] to facilitate the processes of molecular clustering near the membrane. Metadynamics is

an enhanced sampling scheme where the auxiliary Gaussian potentials are imposed along a user-defined order parameter (also known as collective variables, reaction coordinates, etc.) to reconstruct the free energy profile of the process of interest. We have used the $R_g^{system}$ as our order parameter. The middle bead of each molecule (500 beads for 500 molecules) is considered to compute $R_g^{system}$. Once the order parameter reaches a minimal level (Supplementary Fig. 1), a hemispherical cluster is formed. We then run standard Langevin dynamics (without any biasing potential) at different values of Es and Ens to construct the phase diagram. We run metadynamics for 200 million steps to create the hemisphere. Then we relax the system with 300 million steps for each combination of Es and Ens. Then another 200 million steps are executed to sample 20 snapshots for each condition.

**Data analysis.** For metadynamics simulations, we analyzed the time evolution of the order parameter and related potential mean force (PMF) profile. To analyze the physical properties of the cluster, we used the configuration files ("restart" files in LAMMPS) containing information on the coordinates and topology of the system. The topology information was then converted into a network (cluster) to infer the connected components. The molecular connectivity provides the cluster size distribution where a cluster is a molecular network. For example, each of the unconnected molecules (monomers) are clusters of size 1. The fraction of total molecules in the cluster of size $S_i$,

$$f_i = \frac{1}{N_{total}} \sum_i n_i S_i$$

where $n_i$ is the number of clusters of size $S_i$ and $N_{total}$ is the total molecular count. We then define an average measure of the cluster size distribution in the form of ACO or average cluster occupancy (Supplementary Fig. 11),

$$ACO = \sum_i f_i S_i = \frac{1}{N_{total}} \sum_i n_i S_i^2$$

ACO is then normalized by the total molecular count ($N_{total}$). The upper limit of this normalized parameter is now 1. It quantifies the clustering state of the system (Fig. 1C).

The $R_g^{system}$ indicates the compaction or density of the system, which is derived from the particle coordinates (Fig. 1D). To obtain the upper limit of $R_g^{system}$, we turn off all the energy terms (Es = 0, Ens = 0). With only excluded volume interactions in place, we refer to this limit as $R_g^{dispersed}$.

Once the cluster is relaxed for a combination of Es and Ens, we compute the axial radii (Lx, Ly and Lz, Fig. 1B) along the three directions. If a, b, and c are axial lengths, Lx = a/2, Ly = b/2, and Lz = c. Since Lx and Ly are symmetric due to the membrane, we compute a Hemi-Sphericity (Fig. 1E) by taking the ratio of Lz and the average of Lx and Ly,

$$hemiSpericty = \frac{L_z}{L_{xy}}$$

where

$$L_{xy} = \frac{L_x + L_y}{2}$$

To compute the molecular locations within the cluster, firstly, the cluster centroid is computed, and the centroid is projected onto the XY plane (by setting z-component to 0). From this "basal" point, distance of each molecular centroid is calculated along the X, Y, and Z directions ($d_x$, $d_y$, and $d_z$, respectively). The distances are then normalized by the respective axial lengths of the cluster to derive a "normalized radial distance" which highlights the central vs peripheral locations of the molecules (Fig. 1F).

**Data representations.** To display box-plot type distribution (e.g., Fig. 1E), the box represents the interquartile range (IQR) from the 25th percentile (Q1) to the 75th percentile (Q3). The line inside the box indicates the median (50th percentile) of the data. The whisker extends from the box to the smallest and largest values within 1.5 times the IQR from Q1 and Q3, respectively. Data points outside the range of the whiskers are outliers and are shown as individual points.

## Software

**Moltemplate.** To create the model polymers, we made use of the Moltemplate[44] package which enables the user to create multiple types of chains in a template based manner.

**Packmol.** The polymers are packed inside the simulation volume using the PACKMOL[45] package.

**LAMMPS.** The Langevin dynamics simulations are performed using the LAMMPS software package[42]. We used the "fix COLVARS" functionality[46] to perform metadynamics within LAMMPS. To model reversible inter-sticker bond formation, we used the "fix bond/create/random" module[47].

**OVITO.** We used the OVITO (basic version) software to visualize the particle motions[48].

**Python.** We used custom Python scripts to setup simulations and analyze the data.

We have organized and released input files for simulations and Python code to analyze the data.

## Reaction-diffusion simulations

**Virtual cell software.** We used the Virtual Cell (VCell) software[49,50] to build and simulate the reaction-diffusion models. VCell can simulate a reaction-diffusion system within an arbitrary geometry. The detailed algorithm, including numerical solvers, boundary conditions, etc., can be found here[51].

Briefly, the governing equation is

$$\delta C_i / dt = D_i \nabla^2 C_i + R_i, i = 1, 2, 3, \dots, n$$

where $C_i$ and $D_i$ are the concentration and diffusion constant of the $i^{th}$ species, respectively. The effect of all the reactions on the $i^{th}$ species is captured by the reaction term $R_i$, which depends on the concentrations, $C_1, C_2, C_3, \dots, C_n$. If there are m reactions with rate $\beta_j$ where $j = 1, 2, 3, \dots, m$, then

$$R_i = \sum_j \alpha_{ij} \beta_j$$

$\alpha_{ij}$ is the stoichiometry matrix that represents how many molecules of $i^{th}$ species are consumed or produced during the $j^{th}$ reaction.

**Simulation setup.** Since we are modeling the actin branching near the membrane-associated condensates, we have three components – Arp2/3, G-actin (GA), and F-actin (FA). We have used a lumped kinetic expression (similar to[52]) to model the dendritic nucleation of actin branching, described by Eqs. 1 and 2.

In the single-condensate model, there are two spatial compartments – one is condensate, and the other is cytosol. To resolve and separate out spatial domains, VCell assigns a "membrane" between compartments, which will be called CM or condensate membrane. This

should not be confused with the concept of a "cell membrane" which happens to be the XY plane. Our hemispherical condensate is tethered to the XY plane.

GA and FA diffuse in the cytosol. Arp2/3 ("arp_cm") is initially localized on the CM. By convention, VCell needs reacting species to occupy the same compartment to engage in a reaction. Since Arp2/3 facilitates the GA to FA conversion, we have a translocation process which transfers Arp2/3 from CM to the cytosol ("arp_cyt"). The diffusion of arp_cyt is set to be very low. In that way, FA production still happens near the condensate surface and the initial surface density of Arp2/3 determines the reaction propensity.

For our reference system, diffusion constants (in the unit $\mu m^2/s$), GA = 10, FA = 1, arp_cm = 1, arp_cyt = 0.001.

Dimension of the computational domain = 20, 20, 10 $\mu m$, which is divided into a cytosol and a hemispherical condensate. Radius of the hemisphere = 4 $\mu m$. Hence, volume of cytosol = 3866 $\mu m^3$, surface area of the condensate = 100 $\mu m^2$.

Initial concentrations of the model components, GA = 100 $\mu M$, FA = 0.01 $\mu M$, arp_cm = 1000 molecules/$\mu m^2$, arp_cyt = 0.

**Model availability.** Both the single-condensate (Name: AC_public_OneCondensate_ARP_revised) and four-condensate (Name: AC_public_FourCondensate_ARP_revised) models are publicly available in the VCell database, under the username "Ani". VCell is a freely available (https://vcell.org/run-vcell-software) academic software that offers a graphical interface to define geometry, species, and reactions. One can then scan parameters to investigate different biological scenarios. We also provided the VCell input files (.vcml or virtual cell markup language) in the supplementary source files that can be imported to VCell.

**Data analysis.** To compute the local concentration of F-actin (FA) near the cluster, we consider a hemispherical shell near the cluster surface (Supplementary Fig. 3). The FA within the shell volume amounts to the surface concentration (Fig. 2C) for one-condensate case or local concentration (Fig. 3C) for four-condensate case. While going from one cluster to a four-cluster state, the shell radius is adjusted in such a way that total shell volume remains approximately constant in both cases. With this approach, the FA production at the cluster's local volume becomes comparable for both scenarios.

While dealing with hemispheroidal clusters (HemiSphericity <1), the shape of the shell mirrors the cluster. As mentioned earlier, the radius and height of the hemispheroidal shell is adjusted to keep the local volume approximately the same (Supplementary Fig. 6).

Python codes to analyze the surface and bulk concentrations of F-actin can be found here (https://github.com/achattaraj/Optimal CondensateSize) in the convenient form of a Jupyter Notebook.

**Solution of the ordinary differential equations (ODEs)**
**Statistical distribution of cluster sizes.** To investigate the effects of unequal cluster sizes, we create cluster size distributions (as detailed in Fig. 5A). To demonstrate the workflow, we will consider a simple case of $n = 4$, that is, splitting of one large cluster into four smaller ones. We sampled the four clusters in such a way that the volume is conserved during the process. Let's say, the radius of the single large cluster is R (Volume = $V_{large}$) and the total count of Arp2/3 is A molecules. If all the smaller clusters have the identical size, then $r_{small} = 4^{-0.33} * R \sim 0.63\,R$. Now, if we want to sample the radii from a Gaussian distribution, we draw four radii (r1, r2, r3, r4) from a normal distribution with mean = $r_{small}$ and a CV (coefficient of variation). Total volume of the four clusters,

$$V_{total} = (4/3)*\pi* \sum_i r_i^3$$

To ensure volume conservation, we compute a scale-factor,

$$scale\,factor = \left(\frac{V_{large}}{V_{toal}}\right)^{1/3}$$

We then multiply the scale-factor to radii of the smaller clusters. Now, the cluster volume after the splitting remains unchanged.

To distribute Arp2/3 amongst the smaller clusters, we assume that extent of Arp2/3 per cluster should be proportional to the volume of the cluster. Since cluster volume decides how much NWASP (and Nck + Nephrin) is in it, and Arp2/3 is recruited by NWASP, Arp2/3 partitioning should be decided by the relative volume fractions of the clusters.

Hence, Arp2/3 count in the $i^{th}$ cluster,

$$a_i = \frac{V_i}{V_{total}}$$

Surface density,

$$S_i = \frac{a_i}{4\pi r_i^2}$$

### Reporting summary
Further information on research design is available in the Nature Portfolio Reporting Summary linked to this article.

## Data availability
Source data are provided with this paper. Source data are also available on Github [https://github.com/achattaraj/OptimalCondensateSize/tree/master/Supplementary_data]. Source data are provided with this paper.

## Code availability
All the source code including representative datasets are publicly available on Github [https://github.com/achattaraj/OptimalCondensateSize], and also published in Zenodo [https://doi.org/10.5281/zenodo.12689891][53]. We have used custom Python scripts to solve the ODEs. All the scripts for Fig. 4 and Fig. 5 are available in the GitHub.

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

## Acknowledgements

We are grateful to Srivastav Ranganathan, Junlang Liu, and Sayantan Mondal for numerous fruitful discussions. We thank Prof. Rohit Pappu (Washington University in St. Louis) for suggesting the analysis with a distribution of cluster sizes. A.C. thanks Junlang Liu for useful discussions regarding the analytical solution, Profs. Les Loew and Michael Blinov (University of Connecticut) for all the help with the Virtual Cell software, Prof. Jon Ditlev (University of Toronto) for providing

constructive feedback on the initial manuscript. This work was supported by NIH Grant 5R35GM139571.

## Author contributions
A.C. conceptualized the project, performed simulations, analyzed the data, and wrote the paper. E.I.S. secured the funding, supervised the project, analyzed the data, and wrote the paper.

## Competing interests
The authors declare no competing interests.
