## [Peer Review File · Nature Communications]

Multi-condensate state as a functional strategy to optimize the cell signaling outputREVIEWER COMMENTS

Reviewer #1 (Remarks to the Author):

Phase separation in cells leads to the formation of biomolecular condensates. In contrast with the phase-transition phenomenology encountered in most non-living materials, phase separation in cells often leads to the formation of multiple condensates rather than a single big droplet. The mechanism and molecular interactions favoring the formation of multiple condensates are currently debated in the community.

The present submission studies the functional advantages of having multiple condensates. In particular, the manuscript studies actin polymerization and shows how this process is optimized in the presence of multiple condensates. A simple Reaction-Diffusion (RD) model is presented allowing the authors to make precise predictions about the number of optimal condensates and favorable size distributions.

I have some major remarks about the Langevin simulations (see below). I do not think that these remarks will impact the outcomes of the RD model as the latter does not rely quantitatively on the outcomes of the Langevin simulations.

It should be noted that Langevin simulations do not yield multiple condensates. It is not clear if this is because the model is not sufficiently refined, because of the small simulation boxes employed, or because of the simulation protocol (i.e. the choice of the order parameter employed in Metadynamics). The stoichiometric ratios between different biomolecules should play a major role. The authors may want to further elaborate on this point.

Also, I am not fully convinced about the use of the hemispherical geometry employed in the RD model (see below). In particular, I do not understand the use of a contact angle equal to 90 degrees.

A manuscript fully addressing the points below would be an important contribution to the field and would be interesting to a broad interdisciplinary community. I cannot recommend the publication of the current submission.

Major points:

=About the Langevin simulations.

-I have issues with the algorithm adding/removing bonds between beads interacting through specific interactions. It seems that the current implementation of this algorithm does not satisfy the detailed balance condition. This implies that the simulated ratio between the occurrence of two configurations with and without a given bond is not $\exp[-Ks (R-R_0)^2]$ as one would expect. There are two problems:

1) p_{on} and p_{off} are not a function of R while the detailed balance condition requires

$$p_{on}/p_{off} = \exp[-Ks (R-R_0)^2]$$

Usually, the R dependence is entirely accounted for by only one reaction rate (e.g., p_{off} for slip bonds).

2) It is legitimate (for computational efficiency) to bind two monomers only when found at a distance smaller than R_c . But then, to satisfy the previous equation, unbinding should be allowed only for bound monomers satisfying the same condition.

-The forcefield used in Langevin dynamics comprises both non-specific and specific interactions. It is not clear what is the contribution of these two classes of interaction in driving phase separation.

-I found it very interesting that the authors implemented the reaction-diffusion algorithm in LAMMPS. To my knowledge, it is difficult to design reaction-diffusion algorithms using parallel MD engines. The authors may want to share details and the software (e.g. plugins) about their implementation as it could be very interesting to the community.

=In the Reaction-Diffusion (RD) model, Arp2/3 diffuses slowly around the condensate surface. However, this does not seem consistent with Fig. 1D where Arp2/3 is also found in the cytosol. What outcomes of the Langevin simulations have been used to parametrize the RD model? Only the fact that the condensate is spherical? If this is the case, the following statement in the Discussion 'It also demonstrates how to amalgamate scale-specific insights into a multi-scale picture' is not well motivated. Why Arp2/3 can convert GA only at the surface? (Again, Fig. 1D shows that Arp2/3 is also present in the internal region of the condensate.)

=Hemispherical approximation.

-The shape of the condensate in Fig. 1B does not look consistent with the one in Fig. 1C. In general, the use of a contact angle equal to 90 degrees does not look justified. For instance, by decreasing the number of Nck molecules, one should be capable of reducing the cross-linking between NWASP molecules, therefore reducing the thickness of the condensate (maximal distance between the condensate membrane and the surface) and, in the end, reducing the contact angle. Are the results of the manuscript robust against variations in the contact angle?

=I did not completely grasp the results of Fig. 2C D. I understand that the gap between surface and bulk curves decreases when increasing D_{FA} . However, why does the concentration of the bulk F-actin not increase? (Are the y-axis of the two panels of Fig. 2C identical?)

=The RD model neglects interactions (e.g. steric interactions) between solute molecules. Is this approximation justified? The interface of the condensate could be quite crowded.

=Introduction & Discussions. It is stated that equilibrium theories cannot predict the formation of multiple aggregates. I politely disagree. There are systems in soft matter in which the size of the aggregates is thermodynamically controlled. Size control was achieved in colloidal science a long time ago using long-range repulsive tails. Relevant to bio condensates, more recently, multivalent particles with competing intra-molecular and inter-molecular interactions in the presence of triggering seeds have been used to yield finite-sized aggregates in equilibrium conditions. Similarly, the kinetic mechanism presented in Ref. 7 has also been proposed in theoretical studies of multivalent interactions.

Minor points:

=In the introduction, the authors distinguish between scaffold and client biomolecules. Is this classification relevant to the studied system?

=Metadynamics. I assume that the order parameter in Fig. S1A is calculated using the PMF at time t and is not the value of the order parameter featured by the system at time t (which I think is biased). The explanation of the order parameter given in the caption of Fig. S1A is clearer than the one given in the main text, on pg. 4.

Reviewer #2 (Remarks to the Author):

Combining Langevin dynamics, reaction-diffusion simulation, and dynamical systems theory, this work presents a novel strategy for optimizing the function of multiple condensate states (MCS). Through metadynamics simulation and standard Langevin Dynamics, the study reveals that the cluster comprised of Nephhrin, Nck, NWASP, and Arp2/3 exhibits a hemispherical structure, with Arp2/3 diffusing near the surface of the cluster. Subsequently, a reaction-diffusion model illustrates that Arp2/3 diffusion at the surface of a hemispherical cluster leads to local focal adhesion (FA) accumulation around the clusters, with multi-cluster states resulting in greater global F-actin production. Utilizing systems theoretic analysis based on ordinary differential equations (ODE), the study identifies an optimal scenario wherein F-actin production reaches its maximum due to highly nonlinear production rates. Moreover, by analyzing statistical distributions of cluster sizes, distinct F-actin production profiles are observed, which could potentially correlate with disease phenotypes in cellular systems.

This study exemplifies how multiple computational tools can synergistically contribute to a comprehensive understanding of biological phenomena. It offers compelling insights into the

presence of multiple condensates within living cells. However, several questions arise from this study:

1. The descriptions of Fig2A and 2B require further clarification. It is challenging to understand how the reaction diffusion model was executed and discern the difference between surface and bulk concentrations in these figures.

2. The paper's conclusions appear somewhat unsubstantiated. It is essential to provide additional data to bolster these claims. With two examples, numerous analogous issues are evident throughout the work. Consequently, it is imperative that the authors address this matter with due diligence.

Example 1: the sentence "Fig. 1D shows that only Arp2/3 has a long tail along the normalized radial distance, suggesting a peripheral location. Hence, the cluster made of Nephrin, Nck, NWASP and Arp2/3 is hemi-spherical, and Arp2/3 diffuses near the cluster surface.", this needs more evidence to demonstrate the Arp2/3 diffuses near the cluster surface.

Example 2: the sentence "Fig. 2D reveals a pattern of FA surface enrichment as a function of differential diffusion of FA. With slower FA diffusion, local accumulation of FA becomes more prominent. Hence, when Arp2/3 localizes in the cluster and catalyzes the GA to FA conversion. local enrichment of FA is a kinetic phenomenon which is a direct effect of slower diffusion of polymeric actin compared to its monomeric counterpart." Extracting multiple conclusions solely from the figures proves challenging for readers. More explicit elucidation within the text would facilitate comprehension and enhance the paper's accessibility.

3. By creating a more asymmetric distribution (from gaussian to exponential), a "spiked" response becomes more "sustained". However, there is a limit to this process - creating a fat-tailed distribution like a power-law might triggers a pathological state which simply cannot respond to the external signal. Depending on the spatiotemporal context, different cellular systems can exploit these distributions to achieve specific tasks. Regarding the discussion on asymmetric distribution and its implications, providing detailed examples of cellular systems exploiting such distributions to accomplish specific tasks would enrich the analysis and enhance understanding.

Reviewers' comments are in black, our rebuttals are in red.

We have used the following acronyms throughout the report - LD: Langevin Dynamics, RD: Reaction diffusion, ODE: Ordinary differential equation, Es: specific interaction energy, Ens: Non-specific interaction energy, LJ: Lennard Jones

Reviewer #1 (Remarks to the Author):

Phase separation in cells leads to the formation of biomolecular condensates. In contrast with the phase-transition phenomenology encountered in most non-living materials, phase separation in cells often leads to the formation of multiple condensates rather than a single big droplet. The mechanism and molecular interactions favoring the formation of multiple condensates are currently debated in the community.

The present submission studies the functional advantages of having multiple condensates. In particular, the manuscript studies actin polymerization and shows how this process is optimized in the presence of multiple condensates. A simple Reaction-Diffusion (RD) model is presented allowing the authors to make precise predictions about the number of optimal condensates and favorable size distributions.

We thank the reviewer for their careful reading and constructive feedback. The comments were very helpful in improving the quality of the work.

I have some major remarks about the Langevin simulations (see below). I do not think that these remarks will impact the outcomes of the RD model as the latter does not rely quantitatively on the outcomes of the Langevin simulations.

To accommodate the reviewer's request, we significantly modified and expanded our LD simulations (detailed below).

It should be noted that Langevin simulations do not yield multiple condensates. It is not clear if this is because the model is not sufficiently refined, because of the small simulation boxes employed, or because of the simulation protocol (i.e. the choice of the order parameter employed in Metadynamics).

The LD simulations do not yield multiple condensates because (a) of the box dimension being relatively smaller (b) of the metadynamics simulations which facilitate the clustering by the enhanced sampling method. However, the physical principles underlying the existence of multiple droplets are systemically explored elsewhere (1). In the current study, using LD, we sought to quantify the shape of the condensate and the relative location of molecular types within the condensate. We did not attempt to create multi-condensate scenarios which depend on multiple factors like sticker saturation and relative energetic difference in sticker-sticker and sticker-spacer interactions. The underlying idea is that each droplet (in the n-droplet system) will have similar spatial architectures as the one studied here.

The stoichiometric ratios between different biomolecules should play a major role. The authors may want to further elaborate on this point.

The stoichiometry plays important roles in clustering of multivalent biomolecules. Chattaraj et al. (2) previously explored the role of stoichiometry and valency (amongst another factors) of the same system. We restrict our LD initial conditions to ideal stoichiometry (Nephtin : Nck : NWASP : Arp2/3 = 1 : 3 : 1.5 : 0.75). For this system, there are experimental evidence (3, 4) that suggest the condensate composition being majorly governed the ideal stoichiometry.

Also, I am not fully convinced about the use of the hemispherical geometry employed in the RD model (see below). In particular, I do not understand the use of a contact angle equal to 90 degrees.

In the revised version, we have now explored a wide range of cluster geometry. From our LD simulations, we quantified the HemiSphericity parameter (Fig. 1E) of the cluster which is directly related to the contact angle. By scanning the energy parameters, we extracted the range of HemiSphericity and used those values in our RD (Figs. 2 and 3) and ODE analyses (Fig. 4).

A manuscript fully addressing the points below would be an important contribution to the field and would be interesting to a broad interdisciplinary community. I cannot recommend the publication of the current submission.

We hope that our revised manuscript will address all the concerns. We thank the reviewer for appreciating the potential significance of our work.

Major points:

=About the Langevin simulations.

-I have issues with the algorithm adding/removing bonds between beads interacting through specific interactions. It seems that the current implementation of this algorithm does not satisfy the detailed balance condition. This implies that the simulated ratio between the occurrence of two configurations with and without a given bond is not $\exp[-Ks (R-R_0)^2]$ as one would expect. There are two problems:

1) p_{on} and p_{off} are not a function of R while the detailed balance condition requires

$$p_{on}/p_{off} = \exp[-Ks (R-R_0)^2]$$

Usually, the R dependence is entirely accounted for by only one reaction rate (e.g., p_{off} for slip bonds).

2) It is legitimate (for computational efficiency) to bind two monomers only when found at a distance smaller than R_c . But then, to satisfy the previous equation, unbinding should be allowed only for bound monomers satisfying the same condition.

In the revised version, we have modified the force-field (Sup. Fig. 9 and Method) for sticker-sticker specific interactions. We now use a shifted harmonic potential to model inter-sticker bonds. The idea of a "bond" maintains the single valence nature (which in turn mimics the biochemical specificity) of inter-sticker interactions. With the modified treatment, the bond formation provides energetic gain. In other words, the system's energy drops by E_s once a

sticker pair forms a bond. This modified potential for E_s behaves similarly (Sup. Fig. 9) as the LJ potential (E_{ns}), except the single valence nature for E_s .

Although the inter-sticker binding scheme is akin to Monte Carlo simulations, we set the binding and unbinding probabilities to 1 *once the distance criteria (R_{cut}) is satisfied*. It means that the binding would behave like a diffusion-limited process and the unbinding rate would be inversely proportional to the depth of the specific energy well (E_s). **To avoid confusion, we note that by binding event we mean not the bond length reaching the minimum energy but infinitesimal crossing of the cutoff length (R_{cut}) at which the bond is formally “formed” or “broken”. At this condition $p_{on}=p_{off}=1$ setup is reasonable and consistent with the detailed balance. The rate to break the bond that was initially at the minimum energy will then follow the normal Arrhenius law as outlined by the reviewer while the rate to form the bond by crossing the R_{cut} criterion will be a constant determined by diffusion coefficient. As a result, the frequencies of bonded vs non-bonded stickers should follow the Boltzmann distribution.**

We have updated the relevant method section – “Since both the probabilities (p_{on} , p_{off}) are set to 1, the stochastic factor of inter-sticker binding and unbinding at $R=R_{cut}$ is absent. For a probability < 1 , there is a stochastic factor that determines whether to make or break the bond even when the distance criteria is satisfied. In our case, the bond formation or breakage only depends on the inter-sticker distance. The lifetime of the bond becomes a function of E_s , such that, $\tau_b \propto e^{E_s/kT}$. Indeed, the inter-sticker dissociation events decay exponentially with higher E_s (Supplementary Fig. 10), consistent with an Arrhenius-like rate expression, $Rate \propto e^{-E_s/kT}$.”

-The forcefield used in Langevin dynamics comprises both non-specific and specific interactions. It is not clear what is the contribution of these two classes of interaction in driving phase separation.

We have added full phase diagrams (Figs. 1C and 1D) where we scanned the energy parameters to dissect the relative contributions of specific and non-specific interactions.

-I found it very interesting that the authors implemented the reaction-diffusion algorithm in LAMMPS. To my knowledge, it is difficult to design reaction-diffusion algorithms using parallel MD engines. The authors may want to share details and the software (e.g. plugins) about their implementation as it could be very interesting to the community.

As now highlighted in the software section, “To model reversible inter-sticker bond formation, we used the “fix bond/create/random” module (5).” We also provided the input scripts used to run the LAMMPS simulation.

=In the Reaction-Diffusion (RD) model, Arp2/3 diffuses slowly around the condensate surface. However, this does not seem consistent with Fig. 1D where Arp2/3 is also found in the cytosol. What outcomes of the Langevin simulations have been used to parametrize the RD model? Only the fact that the condensate is spherical? If this is the case, the following statement in the Discussion ‘It also demonstrates how to amalgamate scale-specific insights into a multi-scale picture’ is not well motivated. Why Arp2/3 can convert GA only at the surface? (Again, Fig. 1D shows that Arp2/3 is also present in the internal region of the condensate.)

As opposed to our initial manuscript, we now have characterized the cluster shape (HemiSphericity, Fig. 1E) using LD. We then used those HemiSphericity values to perform our RD simulations (Figs. 2 and 3).

Regarding the cytosolic Arp2/3, we originally mentioned in the introduction – “Arp2/3 complex is a nucleation promoting factor (Ref. 25 in main text) that promotes formation of the branched F-actin network. In resting cells, Arp2/3 remains in an “off” state. Given an extracellular signal, membrane-bound receptor, Nephrin forms cluster and bind to an adapter protein, Nck. Nck recruits NWASP which, in turn, activates the Arp2/3 complex and initiates the F-actin branching. The active Arp2/3 sits on the side of an existing filament (“mother filament”) and creates a branched filament (“daughter filament”) along a 70° angular direction. The extent of branched F-actin network connects the cell’s ability to effectively respond to the external signal.” This off-to-on transition of Arp2/3 happens when it interacts with two NWASP at the same time (6). Experiments also suggest that colocalization of NWASP and Arp2/3 at the condensates promotes local F-actin production (4, 7) as opposed to the bulk phase. Our reaction criteria are motivated by these experimental observations.

Regarding the surface location of Arp2/3, we modified our distance calculations compared to our original submission. Since, for a hemisphere, Z-direction is not symmetric with X or Y, it was somewhat misleading to normalize the radial distance by the same radius of gyration factor along all three directions. In the modified version, we now compute the molecular distances and axial lengths for three directions separately and present the normalized distances along each direction (Fig. 1F and Sup. Fig. 2). The peak of Arp2/3 location (not just a tail) is now towards the periphery. Based on this result, we assumed the surface location of Arp2/3 for our RD and ODE models. There is another intuition behind this assumption. Since actin-polymerization is used to push the cell membrane (during cell movement, for instance), it is convenient to polymerize actin outside the condensate such that freshly produced F-actin can bend the membrane without having to distort the condensate shape.

=Hemispherical approximation.

-The shape of the condensate in Fig. 1B does not look consistent with the one in Fig. 1C.

Figure 1 panels are updated. We have now used more direct measures to quantify the spherical shape of the condensate.

In general, the use of a contact angle equal to 90 degrees does not look justified. For instance, by decreasing the number of Nck molecules, one should be capable of reducing the cross-linking between NWASP molecules, therefore reducing the thickness of the condensate (maximal distance between the condensate membrane and the surface) and, in the end, reducing the contact angle. Are the results of the manuscript robust against variations in the contact angle?

We have now explored the effect of the condensate shape. Creating hemispheres with oblate shape (hemiSphericity < 1 and contact angle < 90 degrees) increases the surface area. Thus, Arp2/3 density at the surface reduces if we consider identical Arp2/3 count. Our understanding of shape dependence is culminated in Figs. 4E and 4F with the help of Equations 5 and 6. Please see appendices 3 and 4 for details. To our satisfaction, the idea of a maximal global F-actin is robust against condensate shape fluctuations.

=I did not completely grasp the results of Fig. 2C D. I understand that the gap between surface and bulk curves decreases when increasing D_{FA} . However, why does the concentration of the bulk F-actin not increase? (Are the y-axis of the two panels of Fig. 2C identical?)

We are not entirely sure about the origin of the question. However, FA production is a function of diffusion coefficients of both GA and FA. To illustrate this further, we added two supplementary figures (Sup. Figs. 3 and 4) to show the concentration profiles along a spatial direction. As visible from Sup. Fig. 4, the peak near cluster surface goes down with higher D_{FA} . That's why, surface concentration goes down with higher D_{FA} . Bulk concentration should mirror the trend as it includes all the volume elements including the surface.

=The RD model neglects interactions (e.g. steric interactions) between solute molecules. Is this approximation justified? The interface of the condensate could be quite crowded.

We agree that the steric effects like crowding could play a role. However, the standard RD treats every component as point particle. This is a trade-off between scalability and accuracy for biophysical simulations. The RD simulations, in its current form, are widely used to infer the spatial effects on chemical reactions. Given the peripheral location of Arp2/3, F-actin production should not be majorly effected by crowding. In that case, the relevant parameter would be the polymerization rate constant (k_1) in Equation 1. Fig. 4C provides a state space where we showed how the critical value of Arp2/3 density ($\alpha_{critical}$) depends on k_1 .

=Introduction & Discussions. It is stated that equilibrium theories cannot predict the formation of multiple aggregates. I politely disagree. There are systems in soft matter in which the size of the aggregates is thermodynamically controlled. Size control was achieved in colloidal science a long time ago using long-range repulsive tails. Relevant to bio condensates, more recently, multivalent particles with competing intra-molecular and inter-molecular interactions in the presence of triggering seeds have been used to yield finite-sized aggregates in equilibrium conditions. Similarly, the kinetic mechanism presented in Ref. 7 has also been proposed in theoretical studies of multivalent interactions.

We have to politely disagree with this remark. In our recent report (1), along with our previous study (8), we showed the metastable nature of the sticker-spacer condensates. They are metastable in a sense that the final outcome depends on the initial condition of the system. As highlighted in Figure 2 of the preprint (1), two clusters composed of 200 chains each cannot merge while 400 uniformly distributed chains coalesce into one single cluster under same energy parameters. This behavior is corroborated by recent in-vitro and in-vivo experiments (9). For a system at equilibrium, this initial condition dependence should not be present. However, there are other proposed mechanisms like active processes or physical barriers that might control the condensate size as well.

Minor points:

=In the introduction, the authors distinguish between scaffold and client biomolecules. Is this classification relevant to the studied system?

There is a lack of consensus in the condensate community regarding the scaffold and client nomenclature. We have used the convention in a very specific sense. Without scaffolds, phase separation does not occur, while clients are not drivers for the same. Due to higher number of binding sites (stickers), Nephrin, Nck and NWASP engage in heterotypic multivalent interactions to produce the condensate and recruit the Arp2/3. The clustering will still take place if we remove Arp2/3 from the system; but that is not true for either Nephrin, or Nck, or NWASP. Hence, we presented our four-component system using the scaffold-client framework. Of course, the results of our study do not depend on this classification in any way. However, the framework may provide a generic functional perspective on how a multi-component condensate might employ a spatially heterogeneous architecture to achieve specific tasks.

=Metadynamics. I assume that the order parameter in Fig. S1A is calculated using the PMF at time t and is not the value of the order parameter featured by the system at time t (which I think is biased). The explanation of the order parameter given in the caption of Fig. S1A is clearer than the one given in the main text, on pg. 4.

Previously, we used an inter-group distance metric as our biasing parameter. In the revised version, we have used R_g^{system} (method section) which is more intuitive to understand. Taking the middle bead as a proxy of the respective chain, we compute the radius of gyration of the system and bias along that order parameter.

Reviewer #2 (Remarks to the Author):

Combining Langevin dynamics, reaction-diffusion simulation, and dynamical systems theory, this work presents a novel strategy for optimizing the function of multiple condensate states (MCS). Through metadynamics simulation and standard Langevin Dynamics, the study reveals that the cluster comprised of Nephrin, Nck, NWASP, and Arp2/3 exhibits a hemispherical structure, with Arp2/3 diffusing near the surface of the cluster. Subsequently, a reaction-diffusion model illustrates that Arp2/3 diffusion at the surface of a hemispherical cluster leads to local focal adhesion (FA) accumulation around the clusters, with multi-cluster states resulting in greater global F-actin production. Utilizing systems theoretic analysis based on ordinary differential equations (ODE), the study identifies an optimal scenario wherein F-actin production reaches its maximum due to highly nonlinear production rates. Moreover, by analyzing statistical distributions of cluster sizes, distinct F-actin production profiles are observed, which could potentially correlate with disease phenotypes in cellular systems.

This study exemplifies how multiple computational tools can synergistically contribute to a comprehensive understanding of biological phenomena. It offers compelling insights into the presence of multiple condensates within living cells.

We sincerely thank the reviewers for their positive assessments of our work.

However, several questions arise from this study:

1. The descriptions of Fig 2A and 2B require further clarification. It is challenging to understand how the reaction diffusion model was executed and discern the difference between surface and bulk concentrations in these figures.

We added multiple supplementary figures (Sup. Figs. 3, 4, 6) to show the concentration profiles along a spatial direction and to highlight the “regions of interest (ROIs)” that are considered while computing the surface concentration of F-actin. Hopefully these illustrations will help to clarify.

2. The paper's conclusions appear somewhat unsubstantiated. It is essential to provide additional data to bolster these claims. With two examples, numerous analogous issues are evident throughout the work. Consequently, it is imperative that the authors address this matter with due diligence.

Example 1: the sentence “Fig. 1D shows that only Arp2/3 has a long tail along the normalized radial distance, suggesting a peripheral location. Hence, the cluster made of Nephryn, Nck, NWASP and Arp2/3 is hemi-spherical, and Arp2/3 diffuses near the cluster surface.”, this needs more evidence to demonstrate the Arp2/3 diffuses near the cluster surface.

In our revised version, we analyzed the system in a range of energy parameters (Figs. 1C – 1F) such that it might not appear as anecdotal. We also updated the molecular distance calculation (Fig. 1F, as described above) to reflect the Arp2/3's peripheral location more clearly.

Example 2: the sentence “Fig. 2D reveals a pattern of FA surface enrichment as a function of differential diffusion of FA. With slower FA diffusion, local accumulation of FA becomes more prominent. Hence, when Arp2/3 localizes in the cluster and catalyzes the GA to FA conversion. local enrichment of FA is a kinetic phenomenon which is a direct effect of slower diffusion of polymeric actin compared to its monomeric counterpart.” Extracting multiple conclusions solely from the figures proves challenging for readers. More explicit elucidation within the text would facilitate comprehension and enhance the paper's accessibility.

The sentence was a concluding one summarizing the finding of Figure 2. We modified it to make it less complex - “Hence, Arp2/3 mediated local production of FA at the cluster surface is a kinetic phenomenon that stems from the slower diffusion of polymeric actin compared to its monomeric counterpart.” With the addition of HemiSphericity trend (Fig. 2E), there are now other results around that point.

3. By creating a more asymmetric distribution (from gaussian to exponential), a “spiked” response becomes more “sustained”. However, there is a limit to this process - creating a fat-tailed distribution like a power-law might triggers a pathological state which simply cannot respond to the external signal. Depending on the spatiotemporal context, different cellular systems can exploit these distributions to achieve specific tasks. Regarding the discussion on asymmetric distribution and its implications, providing detailed examples of cellular systems exploiting such distributions to accomplish specific tasks would enrich the analysis and enhance understanding.

There are not much quantitative experimental data available which systematically probed the effect of cluster size distributions on their function. As we discussed originally, Lee et al. (10) quantified the size distributions of natural and synthetic condensates and concluded that physiological condensates conform to an exponential distribution while the pathological aggregates show a power-law distribution. Our analysis predicts that, for the system considered, exponential size distribution produces sustained F-actin response while the power-law like distribution fails to do so.

We added another relevant observation towards the end of our discussion. For enzyme containing synthetic condensates, Gil-Garcia et al. (11) showed that smaller condensates produced faster reaction rates than the larger ones. This observation is consistent with our prediction despite the differences in the systems investigated in (11) and by us.

References

1. Chattaraj A, Shakhnovich EI. Separation of sticker-spacer energetics governs the coalescence of metastable biomolecular condensates. *bioRxiv*. 2023:2023.10.03.560747.
2. Chattaraj A, Youngstrom M, Loew LM. The Interplay of Structural and Cellular Biophysics Controls Clustering of Multivalent Molecules. *Biophys J*. 2019;116(3):560-72.
3. Ditlev JA, Michalski PJ, Huber G, Rivera GM, Mohler WA, Loew LM, et al. Stoichiometry of Nck-dependent actin polymerization in living cells. *The Journal of Cell Biology*. 2012;197:643-58.
4. Case LB, Zhang X, Ditlev JA, Rosen MK. Stoichiometry controls activity of phase-separated clusters of actin signaling proteins. *Science (New York, NY)*. 2019;363(6431):1093-7.
5. de Buyl P, Nies E. A parallel algorithm for step- and chain-growth polymerization in molecular dynamics. *J Chem Phys*. 2015;142(13):134102.
6. Padrick SB, Doolittle LK, Brautigam CA, King DS, Rosen MK. Arp2/3 complex is bound and activated by two WASP proteins. *Proceedings of the National Academy of Sciences*. 2011;108(33):E472-E9.
7. Banjade S, Rosen MK. Phase transitions of multivalent proteins can promote clustering of membrane receptors. *Elife*. 2014;3.
8. Ranganathan S, Shakhnovich EI. Dynamic metastable long-living droplets formed by sticker-spacer proteins. *Elife*. 2020;9.
9. Lin AZ, Ruff KM, Dar F, Jaliha A, King MR, Lalmansingh JM, et al. Dynamical control enables the formation of demixed biomolecular condensates. *Nature Communications*. 2023;14(1):7678.
10. Lee DSW, Choi C-H, Sanders DW, Beckers L, Riback JA, Brangwynne CP, et al. Size distributions of intracellular condensates reflect competition between coalescence and nucleation. *Nature Physics*. 2023;19(4):586-96.
11. Gil-Garcia M, Benítez-Mateos AI, Papp M, Stoffel F, Morelli C, Normak K, et al. Local environment in biomolecular condensates modulates enzymatic activity across length scales. *Nat Commun*. 2024;15(1):3322.

REVIEWER COMMENTS

Reviewer #1 (Remarks to the Author):

The authors extensively worked on and improved the first version of the manuscript. Some points of concern are still present.

Reaction Dynamics.

Traditionally, binding between reacting units has been encoded in the force field (a) or by using hybrid methods, the latter adding/removing bonds using MC schemes (b). In (a), it is usually difficult to enforce valence-limited interactions. In the past, this has been done by using, for instance, flanking/inert beads (tethered to the reacting units) sterically hampering the formation of trimers. On the other hand, the inconvenience of (b) is that, at a given distance, two reacting units may or may not be bound (the Euclidean distance is not a sufficient condition to query the presence of a bond). This feature may look unphysical.

The authors employed a 'mixed' method in which reactions happen with bead diffusing through a cut-off distance (like in a) while valency control is enforced implicitly (similarly to b). This can reproduce the expected dynamics at the two-body level. However, I am still convinced that it breaks the detailed balance at the three-particle level because of the possibility of particle 3 being in the reach of particle 1, the latter forming a bond with particle 2. Specifically, let fix the position of particle 1 in $x=0$ and let evolve particle 2 (3) in $[-2R_c,0]$ ($[0,2R_c]$). Let us also assume that 2 and 1 are bound. As soon as particle 2 stops interacting with 1, particle 1 will instantaneously make a bond with particle 3 (if $x_3 < R_c$). However, particle 3 will be found at a position that is not distributed as $\exp[-E_s(x_3)]$, given that particle 3 is uniformly distributed when 2 is bound to 1. Particle 3 may not have the time to thermalize (according to $\exp[-E_s(x_3)]$) as it may diffuse away from 1 in a finite time. Therefore, at a sufficiently high rate of bond swapping (1-2 \leftrightarrow 1-3), the distribution of x_3 (x_2) conditional on particle 3 (2) being bound to particle 1 is not distributed as $\exp[-E_s(x_3)]$ ($\exp[-E_s(x_2)]$), as one would expect.

On a more physical basis, it should be observed that biochemical association is usually smaller (e.g., by a couple of orders of magnitude in DNA hybridization) than the diffusion-limited association. Association rates may have an impact on the emerging structure and therefore functionalities of a condensate (as shown by the past work of the authors). In the future, it will be important to have a reaction simulation method that allows tuning the reaction rate.

HemiSphericity.

In the new analysis of the condensates' shape, the authors report on how the surface of the condensates is not a spherical cap. How can one justify a liquid interface with a curvature which is not constant? Is this due to non-equilibrium effects (e.g. due to slow unbinding rates) or because of the presence of bending terms? The first case may look plausible given that at the transition the condensates become spherical. In both cases, I think that the degree of hemisphericity should be a function of the size of the condensate.

Missing important literature.

I am surprised by the response of the authors to my last major point 'Introduction & Discussion'. For instance, in Ref. 7 they already acknowledged literature in colloidal science that anticipated the effect of slow unbinding rates on the assembly properties. I do not understand why that is no longer the case.

Reviewer #2 (Remarks to the Author):

Multiple supplementary figures (Sup. Figs. 3, 4, 6) and figures (Figs. 1C – 1F and Fig. 2E) were added to address comment. The author have addressed all Comments.

Reviewer's comments are in black, our rebuttals are in red. Referred sections from the main text are highlighted in yellow.

Reviewer #1 (Remarks to the Author):

The authors extensively worked on and improved the first version of the manuscript. Some points of concern are still present.

We thank the reviewer for recognizing the improvement of the manuscript over the first version.

Reaction Dynamics.

Traditionally, binding between reacting units has been encoded in the force field (a) or by using hybrid methods, the latter adding/removing bonds using MC schemes (b). In (a), it is usually difficult to enforce valence-limited interactions. In the past, this has been done by using, for instance, flanking/inert beads (tethered to the reacting units) sterically hampering the formation of trimers. On the other hand, the inconvenience of (b) is that, at a given distance, two reacting units may or may not be bound (the Euclidean distance is not a sufficient condition to query the presence of a bond). This feature may look unphysical.

The authors employed a 'mixed' method in which reactions happen with bead diffusing through a cut-off distance (like in a) while valency control is enforced implicitly (similarly to b). This can reproduce the expected dynamics at the two-body level. However, I am still convinced that it breaks the detailed balance at the three-particle level because of the possibility of particle 3 being in the reach of particle 1, the latter forming a bond with particle 2. Specifically, let fix the position of particle 1 in $x=0$ and let evolve particle 2 (3) in $[-2R_c,0]$ ($[0,2R_c]$). Let us also assume that 2 and 1 are bound. As soon as particle 2 stops interacting with 1, particle 1 will instantaneously make a bond with particle 3 (if $x_3 < R_c$). However, particle 3 will be found at a position that is not distributed as $\exp[-E_s(x_3)]$, given that particle 3 is uniformly distributed when 2 is bound to 1. Particle 3 may not have the time to thermalize (according to $\exp[-E_s(x_3)]$) as it may diffuse away from 1 in a finite time. Therefore, at a sufficiently high rate of bond swapping ($1-2 \rightleftharpoons 1-3$), the distribution of x_3 (x_2) conditional on particle 3 (2) being bound to particle 1 is not distributed as $\exp[-E_s(x_3)]$ ($\exp[-E_s(x_2)]$), as one would expect.

On a three-particle level, many physical laws may break down and produce counter-intuitive results. The hypothetical scenario depicted by the reviewer is very unlikely for our model as it requires very high frequency of bond swapping. In reality, one pair of bonded stickers does not

engage with their cognate stickers even if they are within the cutoff distance. For a (inter-sticker) bond, if the timescale of vibrational relaxation is short enough compared to the reversible bond formation dynamics, the detailed balance should be obeyed. In our revised version, we provided evidence (Supplementary Fig. 10) that shows an Arrhenius-like rate of inter-sticker dissociation. It confirms that the Boltzmann distribution is obeyed on two-particle level dynamics, since the association rate is a number determined by the particle diffusions and dissociation is a process that requires overcoming the energy barrier of the sticker-sticker bond well and the Arrhenius rates are indicative that stickers are thermalized in their corresponding wells. Now, due to high frequency of bond swapping between two adjacent sticker pairs (3-particle situation), individual bonds may not get enough time to thermalize. We should notice that this scenario would exemplify a non-equilibrium case where violation of detailed balance is not a bug, but a feature.

To make it clearer, we added a few clarifying sentences in the relevant method section of the main text:

“...Indeed, the inter-sticker dissociation events decay exponentially with higher E_s (Supplementary Fig. 10), consistent with an Arrhenius-like rate expression, $Rate \propto e^{-E_s/kT}$. This also indicates that lifetime of individual bonds is sufficient to ensure thermalization within the harmonic well (whose depth is E_s) such that detailed balance is obeyed. Since the inter-sticker association rate is a number determined by the particle diffusions and dissociation is a process that requires overcoming the energy barrier of the sticker-sticker bond well, the Arrhenius rates (Supplementary Fig. 10) are indicative that stickers are thermalized in their corresponding wells.”

On a more physical basis, it should be observed that biochemical association is usually smaller (e.g., by a couple of orders of magnitude in DNA hybridization) than the diffusion-limited association. Association rates may have an impact on the emerging structure and therefore functionalities of a condensate (as shown by the past work of the authors). In the future, it will be important to have a reaction simulation method that allows tuning the reaction rate.

We agree that biochemical reactions have association rates that are lower than diffusion-limited regime. However, for a Smoluchowski-like formalism, it remains challenging to come up with a numerical solver that could simultaneously produce the equilibrium distribution and the transient dynamics. In an attempt to derive an exact association rate expression, Michalski and Loew (1) showed that there is a trade-off between steady state distribution and transient dynamics. Accuracy of one compromise the other.

In our simulation, we can set lower values ($0 < p < 1$) of p_{on} and p_{off} such that the ratio (p_{on} / p_{off}) is the same. But that again could break the detailed balance due to the probabilistic nature of binding and unbinding upon crossing the cutoff distance; that is, for $p_{on} < 1$, when distance $< R_{cut}$, bond may not form. Similarly, bond may not break for distance $> R_{cut}$ if $p_{off} < 1$.

It is certainly an interesting future avenue to address this problem using theory and computation.

Regarding the kinetic arrest phenomenon (the emerging metastable structures) of sticker-spacer protein condensates, it is the single valent nature of sticker-sticker interactions that trigger the arrest. We can observe this arrest, even when p_{on} and p_{off} are set to 1.

HemiSphericity.

In the new analysis of the condensates' shape, the authors report on how the surface of the condensates is not a spherical cap. How can one justify a liquid interface with a curvature which is not constant? Is this due to non-equilibrium effects (e.g. due to slow unbinding rates) or because of the presence of bending terms? The first case may look plausible given that at the transition the condensates become spherical. In both cases, I think that the degree of hemisphericity should be a function of the size of the condensate.

There are emerging evidences which suggest that biomolecular condensates are network fluids (2) or viscoelastic fluids (3) that can have spatial heterogeneity. For a viscoelastic material, the surface minimization effect of surface tension can be counteracted by elasticity stemming from the inter-sticker bond network. In our system, we have four types of molecules. Fig. 1F shows that the condensate is spatially inhomogeneous where molecular organization is not symmetric along the perpendicular (Z -axis in Fig.1) direction. With more favorable binding energy (deeper E_s), membrane bound receptor Nephryn can facilitate the "spreading" process where the condensate flattens and assumes a hemispheroidal shape.

However, even if the clusters were to assume a spherical cap shape, one can show that the single-to-multiple cluster scaling laws remain qualitatively similar. We have added an additional derivation in the form of Supplementary Text 5.

For a hemiSphericity < 1 , we assumed that the larger and smaller cluster have the same shape (identical value of $K(f)$ in Equations 5 and 6). It is a reasonable approximation given the fact that the shape of a condensate should be determined by the binding energies between the membrane-

bound and cytosolic molecules, which remains independent of the condensate size. However, via large scale Langevin Dynamics simulations, it is possible to titrate up the system size (N) and extract the scaling relation with hemisphericity (hS), such that, $hS \sim N^s$ where s is the scaling exponent. This relationship can be directly used to compute the surface density of Arp2/3 which determines the local production F-actin. From Fig. 4F, it is evident that the trend of Global-F is similar across different shapes. One can infer from these trends that the size-dependent shape fluctuation would not alter the existence of an optimal cluster number, although the number might change.

To reflect these points, we added the following paragraph in the result section -

“...We notice that for constructing the scaling laws (Supplementary Texts 3 and 4), we used an oblate spheroid geometry which does not preserve a constant curvature. In an alternative approach (Supplementary Text 5), one can deform the cluster shape maintaining a spherical surface. This constant curvature construction produces qualitatively similar relationship between surface areas of a perfect hemisphere and a spherical cap. We also note that while computing global F, we assumed all the clusters to have identical shape which is characterized by the $K(f)$ parameter. In other words, for $hS = 0.6$, a large and a small cluster will have a hemisphericity of 0.6. It is a reasonable approximation given the fact that the shape of a condensate should be determined by the binding energies between the membrane-bound and cytosolic molecules, which remains independent of the condensate size.”

Missing important literature.

I am surprised by the response of the authors to my last major point ‘Introduction & Discussion’. For instance, in Ref. 7 they already acknowledged literature in colloidal science that anticipated the effect of slow unbinding rates on the assembly properties. I do not understand why that is no longer the case.

We are still not sure what relevant literatures are referred to here. As we indicated in our original response, our understanding of the metastability underlying the sticker-spacer protein condensates has been consolidated with our recent computational study (4) that extends beyond the original idea proposed in Ranganathan & Shakhnovich 2020. The qualitative nature of spatial assembly is different depending on the initial conditions, a hallmark of metastability. This was observed in recent in-vitro and in-vivo experiments as well (5). It is the separation of energy contributions coming from stickers and spacers that make these condensates prone to

kinetic arrest. However, other mechanisms have been proposed to explain this multi-condensate state which are now included in the discussion -

“It is noteworthy that several physical mechanisms like active processes (6, 7), Pickering agent-like surface adsorption (8), growth dependent slower diffusion (9) etc. have been proposed to explain the condensate size distribution. One or multiple of these mechanisms may be operative in stabilizing the multi-condensate state which can be advantageous for functional point of view.”

The Pickering agent model in Ref. 8 is inspired from the stable colloidal suspensions.

Reviewer #2 (Remarks to the Author):

Multiple supplementary figures (Sup. Figs. 3, 4, 6) and figures (Figs. 1C – 1F and Fig. 2E) were added to address comment. The author have addressed all Comments.

We sincerely thank the reviewer for the positive assessment.

References

1. Michalski PJ, Loew LM. SpringSaLaD: A Spatial, Particle-Based Biochemical Simulation Platform with Excluded Volume. *Biophys J.* 2016;110(3):523-9.
2. Dar F, Cohen SR, Mitrea DM, Phillips AH, Nagy G, Leite WC, et al. Biomolecular condensates form spatially inhomogeneous network fluids. *Nature Communications.* 2024;15(1):3413.
3. Alshareedah I, Moosa MM, Pham M, Potoyan DA, Banerjee PR. Programmable viscoelasticity in protein-RNA condensates with disordered sticker-spacer polypeptides. *Nature Communications.* 2021;12(1):6620.
4. Chattaraj A, Shakhnovich EI. Separation of sticker-spacer energetics governs the coalescence of metastable biomolecular condensates. *bioRxiv.* 2023:2023.10.03.560747.
5. Lin AZ, Ruff KM, Dar F, Jalihal A, King MR, Lalmansingh JM, et al. Dynamical control enables the formation of demixed biomolecular condensates. *Nature Communications.* 2023;14(1):7678.
6. Söding J, Zwicker D, Sohrabi-Jahromi S, Boehning M, Kirschbaum J. Mechanisms for Active Regulation of Biomolecular Condensates. *Trends in cell biology.* 2020;30(1):4-14.
7. Zwicker D, Hyman AA, Jülicher F. Suppression of Ostwald ripening in active emulsions. *Physical review E, Statistical, nonlinear, and soft matter physics.* 2015;92(1):012317.
8. Folkmann AW, Putnam A, Lee CF, Seydoux G. Regulation of biomolecular condensates by interfacial protein clusters. *Science (New York, NY).* 2021;373(6560):1218-24.
9. Snead WT, Jalihal AP, Gerbich TM, Seim I, Hu Z, Gladfelter AS. Membrane surfaces regulate assembly of ribonucleoprotein condensates. *Nat Cell Biol.* 2022;24(4):461-70.